# RedSage: A Cybersecurity Generalist LLM

**Naufal Suryanto[1]**, **Muzammal Naseer[1,4]**[*], **Pengfei Li[1]**, **Syed Talal Wasim[2]**,
**Jinhui Yi[2]**, **Juergen Gall[2]**, **Paolo Ceravolo[3]**, **Ernesto Damiani[3]**

[1]Khalifa University, Abu Dhabi, UAE    [2]University of Bonn, Lamarr Institute for ML and AI, Germany
[3]University of Milan, Italy    [4]University of Western Australia, Australia

## Abstract

Cybersecurity operations demand assistant LLMs that support diverse workflows without exposing sensitive data. Existing solutions either rely on proprietary APIs with privacy risks or on open models lacking domain adaptation. To bridge this gap, we curate 11.8B tokens of cybersecurity-focused continual pretraining data via large-scale web filtering and manual collection of high-quality resources, spanning 28.6K documents across frameworks, offensive techniques, and security tools. Building on this, we design an agentic augmentation pipeline that simulates expert workflows to generate 266K multi-turn cybersecurity samples for supervised fine-tuning. Combined with general open-source LLM data, these resources enable the training of RedSage, an open-source, locally deployable cybersecurity assistant with domain-aware pretraining and post-training. To rigorously evaluate the models, we introduce RedSage-Bench, a benchmark with 30K multiple-choice and 240 open-ended Q&A items covering cybersecurity knowledge, skills, and tool expertise. RedSage is further evaluated on established cybersecurity benchmarks (e.g., CTI-Bench, CyberMetric, SECURE) and general LLM benchmarks to assess broader generalization. At the 8B scale, RedSage achieves consistently better results, surpassing the baseline models by up to +5.59 points on cybersecurity benchmarks and +5.05 points on Open LLM Leaderboard tasks. These findings demonstrate that domain-aware agentic augmentation and pre/post-training can not only enhance cybersecurity-specific expertise but also help to improve general reasoning and instruction-following. Project page: https://risys-lab.github.io/RedSage/.

## 1 Introduction

The rapid evolution of cybersecurity threats has elevated the need for proactive and comprehensive defense strategies, as organizations face increasingly sophisticated attacks and advanced persistent threats (Che Mat et al., 2024). Modern cybersecurity involves a wide range of critical tasks, including threat analysis, incident response, vulnerability management, and security monitoring. However,

---

[*]Project Lead

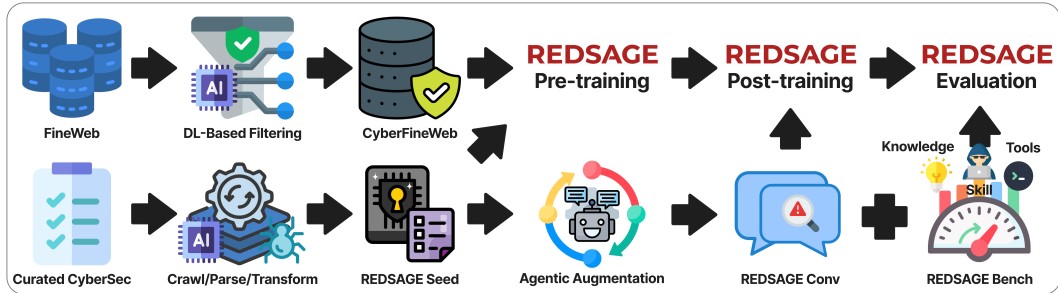

Figure 1: Overview of the RedSage pipeline. RedSage is trained through continual pre-training on cybersecurity-filtered corpora and post-training with curated and augmented conversation data, followed by evaluation on a comprehensive benchmark covering knowledge, skills, and tool expertise.

Table 1: Comparison of cybersecurity LLM benchmarks. Columns indicate knowledge (Know.), skills (Skill), tool proficiency (Tool), and use of quality scoring (Qual.). Size = total samples. Agentic CTF benchmarks (e.g., NYU-CTF, CyBench) are excluded as they are interactive rather than base LLM eval.

| Name | Know. | Skill | Tool | Qual. | Size |
|---|---|---|---|---|---|
| SecEval | ✓ | ✗ | ✗ | ✗ | 2,000 |
| CyberMetric | ✓ | ✗ | ✗ | ✗ | 10,000 |
| CyberBench | ✓ | ✗ | ✗ | ✗ | 80,422 |
| SECURE | ✓ | ✗ | ✗ | ✗ | 4,072 |
| CS-Eval | ✓ | ✗ | ✗ | ✗ | 4,369 |
| SecBench | ✓ | ✗ | ✗ | ✗ | 47,910 |
| CTI-Bench | ✓ | ✓ | ✗ | ✗ | 5,610 |
| CyberSecEval | ✗ | ✓ | ✗ | ✗ | 1,000 |
| RedSage-Bench (Ours) | ✓ | ✓ | ✓ | ✓ | 30,240 |

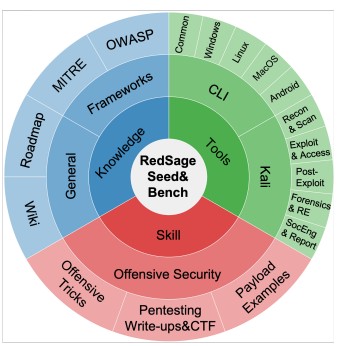

Figure 2: Taxonomy of Red-Sage Seed&Bench dataset. It spans knowledge, practical offensive skills, and tool expertise (CLI and Kali Linux).

the complexity of security tools and the level of expertise required to operate them present significant challenges. These challenges are compounded by a global skills shortage, with research estimating a demand–supply gap of millions of unfilled cybersecurity positions ((ISC)², 2024). Consequently, there is growing momentum to employ cybersecurity-tuned LLMs to augment human analysts.

Recent efforts have produced cybersecurity-trained LLMs, yet most emphasize a single training stage while overlooking others. For instance, some extend pretraining on domain-specific corpora (Kassianik et al., 2025) but apply limited post-training with only 835 samples (Yu et al., 2025) or fewer than 30K cybersecurity-filtered items (Weerawardhena et al., 2025), while others focus on supervised fine-tuning with large cybersecurity Q&A collections without pretraining to strengthen domain knowledge (Deep Hat, 2025). Further, existing cybersecurity benchmarks offer only partial coverage, such as omitting tool proficiency and qualitative evaluation of free-response Q&A beyond simple MCQs (see Table 1 and Fig. 2). Beyond these gaps, most works also do not release their data or pipelines, limiting reproducibility and openness (see Table 2).

To address these gaps, we present RedSage (**R**etrieval-**E**nhanced **D**ata-driven **S**ecurity **A**ssistant **G**uidance and **E**valuation), an open-source LLM tailored for cybersecurity. As illustrated in Fig. 1, RedSage integrates large-scale continual pretraining on cybersecurity-filtered corpora, post-training with curated and agentically augmented datasets, and rigorous evaluation across knowledge, skills, and tool proficiency. Our key contributions are: (1) assembling an 11.8B-token corpus of cybersecurity data for domain-specific continual pretraining, (2) constructing a 266K-sample augmented dataset via an agentic pipeline for supervised fine-tuning, followed by preference alignment with open-source data, (3) introducing RedSage-Bench, a benchmark with 30K MCQs for broad coverage and 240 open-ended Q&A items for quality evaluation across knowledge, skills, and tools, and (4) RedSage, an open 8B model with data and code, achieving state-of-the-art results on established cybersecurity benchmarks while also improving on general benchmarks.

## 2 RELATED WORKS

### 2.1 CYBERSECURITY BENCHMARKS

**General Knowledge.** Several benchmarks assess LLMs' understanding of core cybersecurity concepts via structured Q&A. *SecEval* (Li et al., 2023) includes 2K+ MCQs across nine domains (web, system, application security). *CyberMetric* (Tihanyi et al., 2024) provides 10K MCQs generated with RAG and expert validation, spanning penetration testing and network security. *Cyber-Bench* (Liu et al., 2024) extends beyond MCQs to tasks such as NER, summarization, and classification. *SECURE* (Bhusal et al., 2024) targets Industrial Control Systems with domain-specific MCQs on risk reasoning and vulnerability analysis. *CS-Eval* (Yu et al., 2024) covers 42 subcategories across three cognitive levels (Knowledge, Ability, Application) using MCQs, multi-answer,

Table 2: Comparison of cybersecurity-tuned LLM training datasets. Pretraining and curated columns report token counts (B = billion, M = million). SFT reports the number of supervision samples. ✓= present; ✗= absent; N/R= not reported.

| Name | Pretrain Tokens (B) | Curated Tokens (M) | SFT Samples | Agentic Augmented | Open Data | Open Model |
|---|---|---|---|---|---|---|
| PRIMUS | 2.57 | 191 | 835 | ✗ | ✓ | ✓ |
| Foundation-Sec-8B | 5.10 | ✗ | 28K | ✗ | ✗ | ✓ |
| DeepHat | ✗ | ✗ | >1M | ✗ | ✗ | ✓ |
| Lily-Cybersecurity-7B | ✗ | ✗ | 22K | ✗ | ✗ | ✓ |
| Cyber-DAP | ✗ | 119 | ✗ | ✗ | ✗ | ✗ |
| SecGemini (closed) | N/R | N/R | N/R | ✗ | ✗ | ✗ |
| **Ours (RedSage)** | **11.7** | **850** | **266K** | ✓ | ✓ | ✓ |

Dataset statistics are compiled from official publications, technical reports, and model cards.

T/F, and open-ended items. *SecBench* (Jing et al., 2025) offers 44,823 MCQs and 3,087 SAQs in Chinese and English, capturing both factual recall and logical reasoning.

**Applications and Agentic Tasks.** Application-oriented benchmarks probe reasoning beyond recall. *CTIBench* (Alam et al., 2024) defines four tasks: MCQs, common vulnerabilities and exposures(CVE)-to-common weakness enumeration (CWE) mapping, common vulnerability scoring system (CVSS) prediction, and threat actor attribution in cyber threat intelligence. *CyberSecEval* (Wan et al., 2024) examines model risks across eight areas (e.g., exploit generation, prompt injection). Agentic evaluations such as *NYU-CTF* (Shao et al., 2024) and *CyBench* (Zhang et al., 2025) assess red-team capabilities through capture-the-flag (CTF) challenges in interactive settings.

While these efforts advance evaluation of knowledge and applications, they rarely isolate competence in understanding and operating security tools or systematically assess the quality of free-form responses. As summarized in Table 1, most benchmarks specialize in either knowledge or applications, and even agentic ones lack explicit tool-focused assessment. We address these gaps with RedSage-Bench, which jointly measures knowledge, skills, and tool proficiency (Fig. 2).

## 2.2 CYBERSECURITY DATASETS AND MODELS

**Early Cybersecurity Datasets.** Early domain-specific models such as *CyBERT* (Ranade et al., 2021), *SecureBERT* (Aghaei et al., 2023), and *CTI-BERT* (Park & You, 2023) showed the value of domain-adaptive fine-tuning. However, their datasets were not released and were task-specific.

**Cybersecurity Datasets for LLMs.** With the advent of LLMs, several groups curated cybersecurity-specific corpora. *PRIMUS* (Yu et al., 2025) (Trend Micro) provides 2.75B tokens for continued pretraining, 835 samples for supervised fine-tuning, and reasoning distillation, extending Llama-3.1-8B-Instruct into Llama-Primus-Base and -Merged. *Foundation-Sec-8B* (Kassianik et al., 2025) (Cisco) collects 5.1B tokens via large-scale scraping and filtering, continues pretraining on Llama-3.1-8B-Base, and adds a cybersecurity post-training stage, though its dataset remains closed. Community efforts include *DeepHat* (formerly WhiteRabbitNeo), reportedly trained on 1M+ Q&A pairs for real workflows (Deep Hat, 2025), and *Lily-Cybersecurity*, which fine-tunes Mistral-7B on 22K hand-crafted and lightly refined conversations (Sego Lily Labs, 2024). *Cyber-DAP* (Salahuddin et al., 2025) highlights the effectiveness of smaller curated corpora for continued pretraining, while *SecGemini* (Google Security Blog, 2025) offers a closed model with live threat intelligence but unreleased data. We summarize these datasets in Table 2.

Unlike prior work with limited augmentation, we introduce *agentic augmentation* to transform curated cybersecurity resources into diverse, realistic multi-turn dialogs simulating expert–assistant workflows across knowledge, offensive operations, and tool proficiency for domain-specific fine-tuning. RedSage is, to our knowledge, the only effort combining large-scale continual pretraining, curated data, agentically augmented SFT, and full openness (data, model, and code) (Table 2).

## 3 REDSAGE

We build RedSage through a data-centric pipeline comprising (1) large-scale filtering of cybersecurity text and curation of high-quality resources for continual pretraining, (2) agentic augmentation to create supervised fine-tuning data, and (3) benchmark construction for evaluation (Fig. 3).

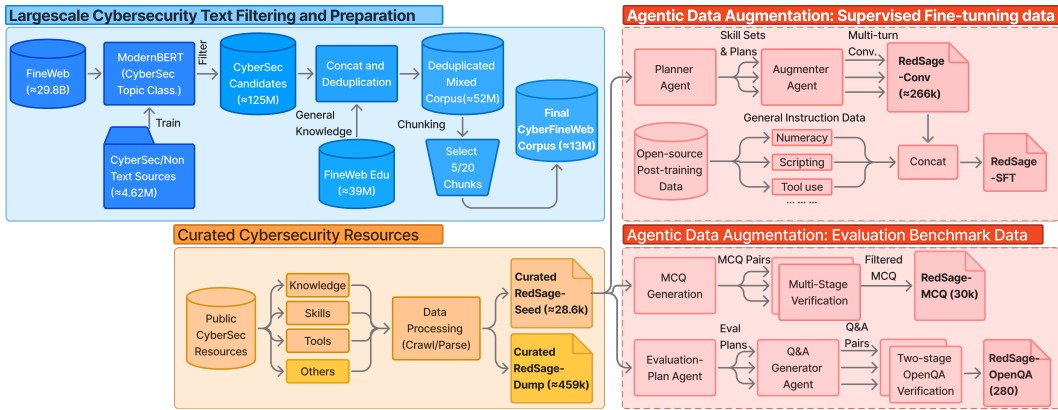

Figure 3: RedSage data pipeline combining large-scale text collection, curated cybersecurity resources, and agentic augmentation for supervised fine-tuning and benchmark generation. *Best viewed in Zoom.*

## 3.1 REDSAGE PRE-TRAINING DATA

**CyberFineWeb.** We construct CyberFineWeb by filtering FineWeb (Penedo et al., 2024a), a cleaned large-scale web corpus aggregated from Common Crawl (2013–2024; ∼15T tokens). To extract cybersecurity content, we fine-tune a binary classification model based on ModernBERT-base (Warner et al., 2025), a state-of-the-art encoder trained on 2T+ tokens. Applying this filter yields a *cybersecurity candidate pool* of ∼125M documents (∼89.8B tokens).

To avoid catastrophic forgetting on general knowledge, we mix CyberFineWeb with general-knowledge samples from FineWeb-Edu (Lozhkov et al., 2024) at a 30% replay ratio. FineWeb-Edu is a 1.3T-token educational subset shown to improve general LLM benchmarks. This strategy follows prior work on replay-based continual learning (Ibrahim et al., 2024; Guo et al., 2025), though unlike dynamic replay, we embed these examples directly into the static corpus. We then apply global near-duplicate removal with MinHash-LSH over the combined data. This yields a deduplicated mixed corpus of ∼52M documents (∼46.8B tokens), while inheriting FineWeb's upstream extensive filtering and PII removal.

Finally, we partition the deduplicated corpus into 20 chronological chunks for sequential training under compute constraints and apply early stopping after 5 chunks to control training cost. This yields the *final CyberFineWeb corpus*: ∼13M documents (∼11.7B tokens) used in our model. Implementation details, including classifier training, deduplication parameters, and datasets statistics, are provided in Appendix A.1.

**RedSage-Seed.** Web-filtered text offers broad coverage, but its reliability is not assured. To provide high-quality content, we curate RedSage-Seed: 28,637 samples (∼0.15B tokens) from publicly available sources organized into three categories: *Knowledge* (well-established cybersecurity frameworks and knowledge bases (MITRE Corporation, 2025a;b;c; The OWASP Foundation, 2025)), *Skills* (penetration-testing write-ups (0xdf, 2025), hacking techniques (HackTricks, 2025), payload examples (swisskyrepo, 2025), and ethical hacking tutorials/blogs (Null Byte, 2025; Chandel, 2025)), and *Tools* (CLI cheat-sheets (tldr pages, 2025), Linux manuals (linux.die.net, 2025), Kali tools (Kali, 2025)). We additionally collect an uncategorized dump of ∼459K documents (∼0.7B tokens) from trusted cybersecurity sources (Appendix A.2) to supply extra pretraining tokens.

To process these resources, we crawl web-based sources and convert them to Markdown using ReaderLM-v2 (Wang et al., 2025), while downloadable resources are parsed directly. This hierarchical Markdown format preserves structure and enables effective chunking for subsequent augmentation stages. Only the categorized seeds are used for augmentation, while both sets support pretraining. Full statistics, categorization, processing steps, and examples are in Appendix A.2.

## 3.2 REDSAGE POST-TRAINING DATA

**Agentic Data Augmentation.** To enable assistants capable of realistic security dialogues, we augment RedSage-Seed into multi-turn conversations using an agentic framework inspired by AgentIn-

Table 3: Statistics of RedSage-Seed (curated pretraining corpus) vs. RedSage-Conv (augmented SFT data) by category. Columns show sample counts, average tokens, and total tokens.

| Category | Seed | | | Conversation | | |
|---|---|---|---|---|---|---|
| | Samples | Avg. Tokens | Tokens (M) | Samples | Avg. Tokens | Tokens (M) |
| Knowledge – General | 6,924 | 2,370 | 16.4 | 67,635 | 1,326 | 89.6 |
| Knowledge – Frameworks | 3,715 | 2,935 | 10.5 | 39,908 | 1,285 | 51.0 |
| Skill – Offensive | 4,032 | 9,478 | 37.8 | 38,870 | 1,345 | 52.3 |
| Tools – CLI | 12,943 | 5,774 | 78.9 | 109,261 | 1,331 | 145.7 |
| Tools – Kali | 1,023 | 6,693 | 6.3 | 10,506 | 1,356 | 14.3 |
| **Total** | **28,637** | **5,231** | **149.8** | **266,180** | **1,326** | **353.0** |
| Cybersecurity Dumps | 459,473 | 1,524 | 700.1 | – | – | – |

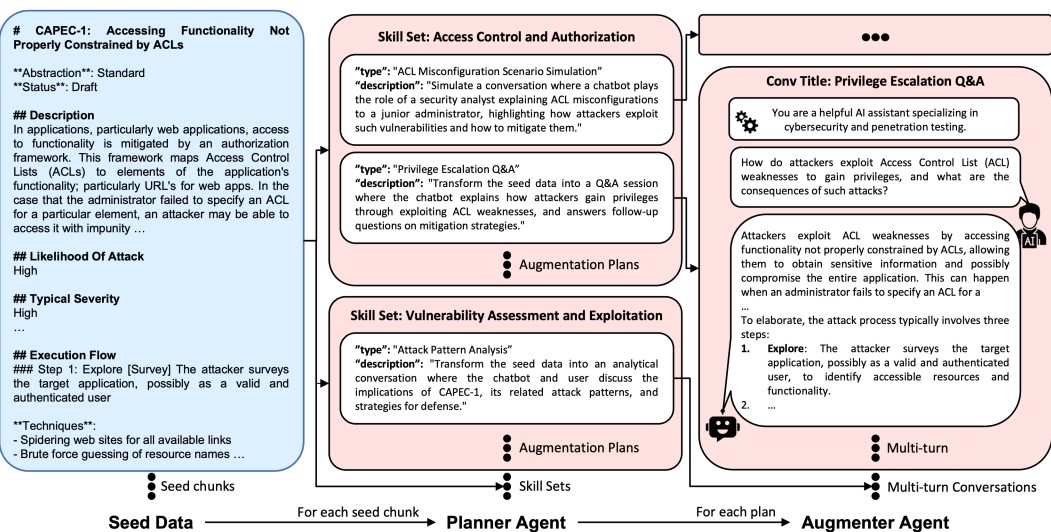

Figure 4: Agentic data augmentation pipeline. Seed data (e.g., CAPEC attack patterns) is processed by the *Planner Agent* into skill sets and augmentation plans, which the *Augmenter Agent* instantiates as grounded, role-based multi-turn cybersecurity dialogues for supervised fine-tuning (SFT).

struct (Mitra et al., 2024). Unlike prior work with fixed skill templates, our *Planner Agent* analyzes each seed data chunk and derives candidate skill sets (e.g., vulnerability analysis, tool-command generation) along with augmentation strategies that describe how the seed is transformed, adapted into a conversational or Q&A format, and enriched with explanations. We enforce guidelines on relevance, diversity, creativity, detail, and formatting. The *Augmenter Agent* then instantiates each plan into realistic, role-based multi-turn dialogues grounded in the seed data. This pipeline scales efficiently, producing multiple dialogues per skill set and filtering outputs for format validity, consistency, and topical relevance. Overall, it yields RedSage-Conv with ∼266K multi-turn conversations (∼352M tokens), expanding total samples by 9.2× and tokens by 2.3× across knowledge, skills, and tools while preserving technical depth (Tab. 3). Fig. 4 illustrates the augmentation pipeline, while detailed statistics, prompts, and examples are provided in Appendix A.3.

**General instruction integration.** While domain-specific conversations ground the assistant in cybersecurity, effective models must also handle broader instruction-following tasks. We therefore complement RedSage-Conv with curated post-training SFT data from SmolLM3 (Bakouch et al., 2025)[1], focusing on its non-reasoning subset. This corpus adds coverage of summarization, numeracy, data interpretation, temporal and unit reasoning, commonsense knowledge, step-by-step planning, technical writing, scripting, and general tool use. The combination of cybersecurity-specific and general instruction data yields a high-quality post-training corpus, enabling a cybersecurity assistant that performs specialized tasks while retaining broad capabilities.

---

[1]General SFT datasets: HuggingFaceTB/smoltalk2

### 3.3 REDSAGE BENCHMARK

**Multiple-choice Q&A generation.** We derive MCQs from RedSage-Seed as follows: for each seed item, a strong open instruction-tuned LLM [2] generates several MCQs under guidelines: items are self-contained and closed-book, target stable domain facts/procedures, follow a four-option format with three plausible distractors, and satisfy diversity and formatting constraints.

**Open-ended Q&A generation.** We extend RedSage-Seed into open-ended Q&A using an agentic augmentation framework with two stages: (1) an *Evaluation-Planner* analyzes seed artifacts and proposes realistic evaluation types with instruction templates and answer guidelines; (2) a *Question-Answer Generator* instantiates each plan into a self-contained open-ended Q&A with a natural-language prompt and a reference answer. All open-ended Q&A are grounded in the seed data and scored with a reference-based LLM-as-judge rubric that evaluates both factual correctness (True/-False) and answer quality (0–10) across helpfulness, relevance, depth, and level of detail.

**Multi-stage verification.** For MCQs, we apply a two-stage pipeline: *Stage 1 (structural validity)* uses a verifier LLM [2] with a checklist on format, correctness, distractors, topical relevance, and consistency, filtering items by pass/fail; *Stage 2 (quality scoring)* then applies the same verifier LLM [2] to assign each remaining item a score $s \in [0, 10]$ for clarity, correctness, and assessment value. In both stages, we use chain-of-thought prompting so the verifier explicitly reasons through each checklist criterion before issuing a pass/fail label or score, yielding judgments that more closely follow our rubric. We then select the pairs where $s > 8$ and apply quota-aware random sampling to ensure taxonomic balance, yielding 30,000 MCQ–answer pairs evenly split across knowledge, skills, and tools. For open-ended Q&A, we directly perform LLM-based quality scoring in *Stage 2* followed by human verification, selecting 240 high-quality pairs evenly distributed across categories.

**Human quality control.** Across all verification stages, we iteratively refined prompts and manually inspected sampled outputs until the verifier consistently aligned with our criteria. We observe that chain-of-thought prompting plays a significant role in producing more precise judgments. For the large-scale MCQ benchmark, random audits confirmed that items passing the final stages met both Stage 1 and Stage 2 requirements. For open-ended Q&A, we retain only human-verified items.

**Data decontamination.** We apply an additional filtering and deduplication step to prevent unintended overlap between our benchmark datasets and augmented post-training data, despite their being generated through different pipelines and output formats. Specifically, we remove any synthetic post-training instance whose query has a semantic similarity above 0.9 to a benchmark question. This eliminates 2.96% of data relative to the benchmark size (0.31% of the full training corpus) and helps ensure that evaluation remains free of training leakage.

Implementation details, intermediate outputs, prompt templates, and qualitative examples are provided in Appendix A.4, and the full evaluation protocol is described in Appendix C.2.

### 3.4 REDSAGE TRAINING

We build RedSage using the Axolotl framework (Axolotl team, 2023), with continued pretraining of the open-source base model, Qwen3-8B-Base (Yang et al., 2025), on cybersecurity corpora, followed by post-training through supervised fine-tuning on augmented conversations and preference alignment. We illustrate training stages in Fig. 5 with further training details, including exact hyperparameters, estimated training time, and computational cost analysis in Appendix B.

**Training setup.** For continued pretraining (CPT), we first train on the CyberFineWeb corpus, followed by training on RedSage-Seed (Sec. 3.1). We run a single epoch with distributed optimization on 32×A100-64GB GPUs (global batch size 1024), using DeepSpeed ZeRO Stage 3, the AdamW optimizer, and a fixed learning rate of $2.5 \times 10^{-6}$ with linear warmup. After pre-training, we further fine-tune our base model on RedSage-Conv and general SFT data (Sec. 3.2) with two epochs using a cosine learning rate schedule. We apply direct preference optimization (DPO) (Rafailov et al., 2023) with open-source Tulu 3 8B Preference Mixture dataset (Lambert et al., 2025) using original hyperparameters.

---

[2]Teacher and Verifier LLM: meta-llama/Llama-3.3-70B-Instruct, Qwen/Qwen2.5-72B-Instruct

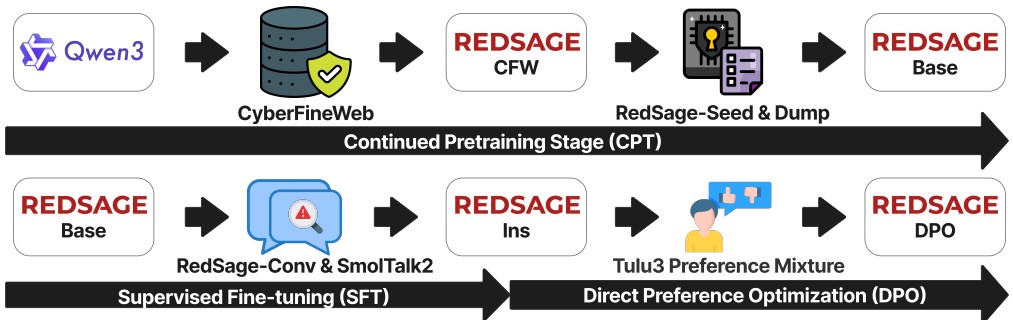

Figure 5: RedSage training pipeline. We first continue pretraining the Qwen3 base model on CyberFineWeb to obtain RedSage-CFW, followed by RedSage-Seed and RedSage-Dump to produce RedSage-Base. We then perform supervised fine-tuning using RedSage-Conv and SmolTalk2 (Bakouch et al., 2025) data, and finalize the model with Direct Preference Optimization using the Tulu3 Preference Mixture (Lambert et al., 2025).

## 4  EXPERIMENTS AND RESULTS

We evaluate the performance of our cybersecurity-tuned LLM on (1) our own benchmark (Sec. 3.3), (2) related cybersecurity benchmarks, and (3) general LLM benchmarks.

**Evaluation setup.** For replicable results, we implement and evaluate RedSage-Bench and prior cybersecurity benchmarks in HuggingFace `lighteval` (Habib et al., 2023). MCQ benchmarks are scored with normalized log-likelihood accuracy over answer options, while instruction-tuned models and structured output tasks use prefix exact match or regex matching on greedy decoding outputs (temperature=0). Details for each task are provided in Appendix C.1.

**Baseline methods.** We evaluate RedSage against both open general-purpose and cybersecurity-tuned LLMs. General-purpose baselines include Llama-3.1-8B (Grattafiori et al., 2024) and Qwen3-8B (Yang et al., 2025), while specialized baselines include Llama-Primus (Base, Merged) (Yu et al., 2025), Foundation-Sec (Base, Ins) (Kassianik et al., 2025; Weerawardhena et al., 2025), Lily-Cybersecurity-7B-v0.2 (Sego Lily Labs, 2024), and DeepHat-V1-7B (Deep Hat, 2025). We also include Qwen3-32B and GPT-5 (OpenAI, 2025) to compare against larger-capacity and proprietary general-purpose models. Base models are evaluated with text completion, instruction-tuned ones with official prompt templates, and we ran hybrid model in non-reasoning mode for fairness.

Our RedSage variants include three base models: **RedSage-8B-CFW** (CyberFineWeb only), **RedSage-8B-Seed** (Seed only), and **RedSage-8B-Base** (CyberFineWeb followed by Seed). We further derive instruction-tuned variants: **RedSage-8B-Ins** (instruction-tuned from Base) and the final **RedSage-8B-DPO**, which combines all data and applies DPO alignment (see Fig. 5). An additional larger-model scaling experiment is presented in Appendix D.1, where partial RedSage data improves a Qwen3-32B model via lightweight QLoRA fine-tuning, demonstrating that our curation pipeline transfers effectively to higher-capacity LLMs.

### 4.1  EVALUATION RESULTS ON REDSAGE-BENCH

**Results on RedSage-Bench.** For MCQs, both base and instruction-tuned models are tested in the 0-shot setting, with Tab. 4 showing that all RedSage variants outperform baselines across categories. For open-ended Q&A, we evaluate instruction-tuned models using an LLM-as-Judge rubric to assess both factual correctness and answer quality (Sec. 3.3). As shown in Fig. 6, RedSage achieves not only high accuracy but also the best answer quality across categories. More detailed results and qualitative examples illustrating model predictions are provided in Appendix C.2.

**Open-ended QA Analysis.** RedSage-8B-DPO achieves the best performance (Fig. 6), surpassing the second-best model (Qwen3-8B) by +7% absolute mean correctness and +0.07 in mean quality score. RedSage-8B-Ins attains similar correctness to Qwen3-8B but lags in answer quality (6.43), underscoring the role of preference alignment in producing not only accurate but also helpful responses. The remaining models fall substantially behind, with mean correctness ranging from 51%

Table 4: RedSage-MCQ (0-shot). Values are accuracy (%). Abbreviations: Gen = General, Frm = Frameworks, Off = Offensive Skills, CLI = Command-line Tools, Kali = Kali Tools. Bold numbers indicate the best result of 8B models; underlined numbers indicate the second best.

| Model Name | Macro | Knowledge | | Skill | Tools | |
|---|---|---|---|---|---|---|
| | Acc | Gen | Frm | Off | CLI | Kali |
| *Base Model Evaluation (Text Completion)* | | | | | | |
| Llama-3.1-8B | 78.02 | 77.42 | 75.26 | 82.78 | 77.78 | 72.12 |
| Foundation-Sec-8B | 78.51 | 76.82 | 79.10 | 83.68 | 76.64 | 71.14 |
| Qwen3-8B-Base | 84.24 | 83.08 | 81.94 | 88.23 | 85.08 | 78.86 |
| **RedSage-8B-CFW** | 84.86 | 83.62 | 83.30 | 88.81 | 85.30 | 79.32 |
| **RedSage-8B-Seed** | **85.21** | **83.64** | 84.56 | **88.82** | **85.50** | **79.90** |
| **RedSage-8B-Base** | 85.05 | 83.12 | **84.94** | 88.72 | 85.44 | 79.36 |
| *Instruct Model Evaluation (w/ Chat Template)* | | | | | | |
| Lily-Cybersecurity-7B-v0.2 | 71.19 | 68.78 | 67.44 | 76.61 | 71.44 | 66.26 |
| Llama-Primus-Merged | 74.81 | 74.34 | 72.34 | 79.31 | 74.74 | 68.82 |
| Foundation-Sec-8B-Instruct | 76.12 | 74.50 | 77.10 | 80.91 | 74.98 | 68.30 |
| Llama-Primus-Base | 77.02 | 76.78 | 74.10 | 80.87 | 76.78 | 72.72 |
| Llama-3.1-8B-Instruct | 77.05 | 76.06 | 73.30 | 80.90 | 78.72 | 72.40 |
| DeepHat-V1-7B | 80.18 | 77.26 | 76.90 | 85.07 | 81.94 | 74.82 |
| Qwen3-8B | 81.85 | 80.46 | 78.82 | 86.16 | 83.92 | 75.56 |
| **RedSage-8B-Ins** | **85.73** | **84.20** | **84.98** | **89.06** | **86.80** | **80.30** |
| **RedSage-8B-DPO** | 84.83 | 82.48 | 83.80 | 88.54 | 86.30 | 79.30 |
| *Larger Instruct & Proprietary Model Evaluation (w/ Chat Template)* | | | | | | |
| Qwen3-32B | 85.40 | 84.08 | 82.32 | 89.00 | 87.60 | 80.40 |
| GPT-5 | 88.68 | 88.74 | 86.54 | 91.43 | 90.80 | 83.14 |

**MCQ Analysis.** Qwen3-8B-Base, trained on 36T tokens, is the strongest external 8B baseline (84.24) and even outperforms Foundation-Sec-8B. underscoring the importance of selecting a strong base model. Building on it with CPT, RedSage gains up to +0.97 macro-accuracy points, with the largest improvements in *Frameworks* (+3.00) and *Kali* (+1.04). RedSage-8B-Seed achieves the best base result (85.21), demonstrating better alignment with the curated Seed data. Among instruction-tuned models, RedSage avoids the accuracy drop and exceeds Qwen3 by +2.98 (DPO) to +3.88 (Ins). DPO on *general data* slightly lowers accuracy but stays well above baselines. Interestingly, RedSage-Ins surpasses Qwen3-32B on average despite its smaller size. These results highlight that our domain-aware CPT and SFT enhance robustness across cybersecurity knowledge, skills, and tools.

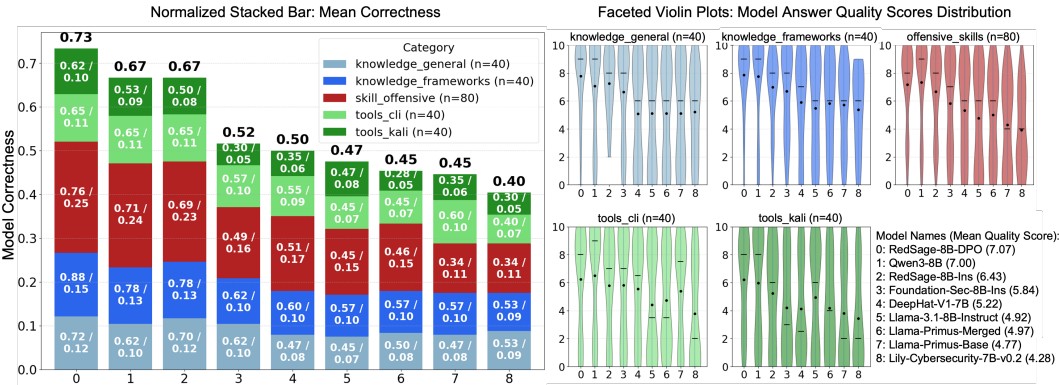

Figure 6: **RedSage open-ended QA evaluation**. Left: normalized stacked bar charts of mean correctness by category (0-1); segment labels show the mean and its relative contribution. Models are ordered by overall mean correctness. Right: faceted violin plots of LLM-as-Judge quality scores (0–10) by category, showing score distributions across models. Black dots indicate means; horizontal lines indicate medians. *Best view in Zoom.*

to 40% and quality scores from 5.84 to 4.28, highlighting a significant gap from the top three. The faceted violin plots further reveal category difficulty: knowledge tasks exhibit higher and tighter distributions, skill tasks lie in the middle range, and tool-use tasks show lower medians with heavy tails, pinpointing tool expertise as the primary challenge. These findings demonstrate the value of our benchmark for assessing cybersecurity capabilities in free-form answers.

## 4.2 EVALUATION RESULTS ON CYBERSECURITY BENCHMARKS

**Results on Cybersecurity Benchmarks.** We assess generalization on multiple established benchmarks in Tab. 5. For CyberMetric (CyMtc) (Tihanyi et al., 2024), we evaluate all models using the 500 human-verified MCQs. We select English (En) MCQs from SecBench (ScBen) (Jing et al., 2025). We also include MCQs related to the Computer Security (CSec) from MMLU (Hendrycks et al., 2021). For SECURE (Bhusal et al., 2024), we evaluate models using the MCQs types covering MAET, CWET, and KCV. Further, we evaluate all models on CTI-Bench (Alam et al., 2024) (MCQ,

Root Cause Mapping (RCM)), and SecEval (ScEva) (Li et al., 2023) (MCQ). We provide further details about each benchmark and metrics in Appendix C.3. Base models are evaluated with 5-shot prompting, and instruction-tuned models in 0-shot.

Table 5: Benchmark results for Base and Instruct Models. Values are Accuracy (%). Rows are sorted by mean performance. Best results for 8B models are in bold, second-best are underlined.

| Model Name | Mean | CTI-Bench | | CyMtc | MMLU | ScBen | ScEva | SECURE | | |
|---|---|---|---|---|---|---|---|---|---|---|
| | | MCQ | RCM | 500 | CSec | En | MCQ | CWET | KCV | MAET |
| *Base Model Evaluation (5-shot)* | | | | | | | | | | |
| Llama-3.1-8B | 75.44 | 61.12 | 65.80 | 84.20 | 83.00 | 72.80 | 54.27 | 86.34 | 83.73 | 87.72 |
| Foundation-Sec-8B | 76.90 | 62.40 | 75.40 | 86.60 | 80.00 | 69.86 | 55.64 | 88.01 | 84.38 | 89.78 |
| Qwen3-8B-Base | 80.81 | 68.80 | 63.50 | 92.00 | 83.00 | 82.84 | 75.60 | 92.70 | 75.05 | 93.81 |
| **RedSage-8B-CFW** | 82.66 | 68.40 | 67.60 | **93.80** | 86.00 | **83.62** | **76.10** | **93.33** | 81.34 | 93.72 |
| **RedSage-8B-Seed** | 84.45 | 70.80 | **78.60** | 92.20 | **88.00** | 81.61 | 75.96 | 93.12 | 85.47 | **94.28** |
| **RedSage-8B-Base** | **84.56** | **71.04** | 78.40 | 92.60 | 87.00 | 81.76 | 75.83 | 93.22 | **87.20** | 94.00 |
| *Instruct Model Evaluation (0-shot)* | | | | | | | | | | |
| Lily-Cybersecurity-7B-v0.2 | 55.74 | 30.04 | 43.60 | 65.20 | 68.00 | 57.65 | 39.72 | 72.99 | 49.67 | 74.79 |
| Llama-3.1-8B-Instruct | 68.52 | 58.24 | 58.30 | 82.80 | 72.00 | 59.66 | 35.37 | 84.98 | 82.86 | 82.47 |
| Llama-Primus-Merged | 71.23 | 55.92 | 68.50 | 83.80 | 76.00 | 64.91 | 39.31 | 86.13 | 82.65 | 83.88 |
| Llama-Primus-Base | 71.69 | 52.32 | 68.50 | 83.80 | **79.00** | 63.68 | 61.15 | 88.01 | 65.08 | 83.69 |
| DeepHat-V1-7B | 75.44 | 62.08 | 68.20 | 86.00 | 74.00 | 70.63 | 56.65 | 87.07 | 86.77 | 87.54 |
| Foundation-Sec-8B-Instruct | 75.44 | 63.24 | 69.40 | 83.00 | 76.00 | 68.78 | 65.46 | 85.82 | 82.00 | 85.29 |
| Qwen3-8B | 75.71 | 62.76 | 54.00 | 88.60 | 76.00 | 73.26 | 65.46 | 88.11 | **87.42** | 85.75 |
| **RedSage-8B-Ins** | **81.30** | 70.56 | **76.70** | 89.80 | 78.00 | 79.91 | 72.48 | **91.45** | 81.34 | **91.47** |
| **RedSage-8B-DPO** | 81.10 | **70.84** | 70.60 | **90.00** | **79.00** | **80.06** | **74.22** | 91.35 | 82.86 | 91.00 |
| *Larger Instruct and Proprietary Model Evaluation (0-shot)* | | | | | | | | | | |
| Qwen3-32B | 82.31 | 70.04 | 65.60 | 91.80 | 84.00 | 84.23 | 76.23 | 89.46 | 89.37 | 90.06 |
| GPT-5 | 86.29 | 76.48 | 74.20 | 95.60 | 86.00 | 87.48 | 83.03 | 92.70 | 88.72 | 92.41 |

**Analysis.** Across related cybersecurity benchmarks, RedSage base models improve over Qwen3-8B-Base (80.81%) by up to +3.75 points. CPT with CFW leads on SecBench (83.62), CyMtc (93.80), and CWET (93.33), raising the mean by +1.85. CPT with Seed excels on CTI-RCM (78.60), MMLU-CSec (88.00), and MAET (94.28), lifting the mean by +3.64. Combining both yields the best overall mean (84.56) and top scores on CTI-MCQ (71.04) and KCV (87.20). In the 0-shot instruct setting, RedSage surpasses Qwen3-8B (75.71%) by +5.39 (DPO) to +5.59 (Ins). Except for Lily-Cybersecurity, all domain-tuned baselines outperform Llama-3.1-8B-Instruct, though still lag behind RedSage. Despite having far fewer parameters, RedSage comes close to Qwen3-32B (82.31 mean, only about +1 point higher) and trails GPT-5 (86.29 mean, roughly +5 points higher), highlighting strong efficiency relative to much larger models. These results show that CyberFineWeb and Seed provide complementary strengths, while different post-training strategies specialize across tasks, together setting new state-of-the-art performance in cybersecurity LLM evaluation.

## 4.3 EVALUATION RESULTS ON GENERAL BENCHMARKS

We use benchmarks from the Open LLM Leaderboard in Lighteval, including ARC-Challenge (ARC-C) (Clark et al., 2018), HellaSwag (HSwag) (Zellers et al., 2019), TruthfulQA (TQA) (Lin et al., 2022), MMLU (Hendrycks et al., 2021), WinoGrande (WinoG) (Sakaguchi et al., 2021), GSM8K (Cobbe et al., 2021), and IFEval (Zhou et al., 2023). Results in Tab. 6 show our instruction-tuned models achieve competitive results on general tasks, surpassing baselines by a clear margin. Benchmark configurations and evaluation metrics are provided in Appendix C.4.

**Analysis.** Among base models, Qwen3-8B-Base is strongest overall (70.86) and leads MMLU (78.73) and ARC-C (68.09), while Llama-3.1-8B tops HSwag (82.08) and WinoG (75.30). Red-Sage bases are competitive in mean (69.23–69.58) and achieve task highs, including best GSM8K (82.34, Seed) and second on MMLU (78.63, CFW) and ARC-C (66.72, CFW), where the slight drop may stem from our FineWeb-Edu general-knowledge replay strategy. After instruction tuning, RedSage attains the best overall mean with DPO (74.33) and second with Ins (73.34), setting new highs on ARC-C (71.76, DPO), GSM8K (86.05, Ins), MMLU (77.38, Ins), and leading WinoG (73.64, Ins). Foundation-Sec-8B-Instruct leads HSwag (81.35) and TQA (53.15), and Qwen3-8B

Table 6: Open LLM Leaderboard Benchmarks. All values are accuracy (%). Bold numbers indicate the best result for 8B models and underlined numbers indicate the second best.

| Model Name | Mean | MMLU | ARC-C | GSM8K | HSwag | TQA | WinoG | IFEvl |
|---|---|---|---|---|---|---|---|---|
| *Base Model Evaluation (Mean excludes IFEval)* | | | | | | | | |
| Llama-3.1-8B | 61.15 | 66.31 | 58.19 | 49.05 | **82.08** | 35.98 | **75.30** | — |
| Foundation-Sec-8B | 60.24 | 63.62 | 58.45 | 46.17 | 81.32 | 38.71 | 73.16 | — |
| Qwen3-8B-Base | **70.86** | **78.73** | **68.09** | 81.73 | 79.62 | **43.84** | 73.16 | — |
| **RedSage-8B-CFW** | 69.31 | 78.63 | 66.72 | 81.12 | 79.26 | 38.09 | 72.06 | — |
| **RedSage-8B-Seed** | 69.58 | 78.18 | 65.19 | **82.34** | 77.76 | 42.44 | 71.59 | — |
| **RedSage-8B-Base** | 69.23 | 77.80 | 65.53 | 82.03 | 77.96 | 42.19 | 69.85 | — |
| *Instruct Model Evaluation (Mean includes IFEval)* | | | | | | | | |
| Lily-Cybersecurity-7B-v0.2 | 56.98 | 56.49 | 58.96 | 30.86 | 80.94 | 48.53 | 72.06 | 50.99 |
| Llama-Primus-Base | 64.82 | 65.09 | 51.19 | 71.80 | 79.49 | 44.62 | 72.69 | 68.85 |
| DeepHat-V1-7B | 64.89 | 69.53 | 57.17 | 77.94 | 74.80 | 33.17 | 69.06 | 72.58 |
| Qwen3-8B | 65.92 | 73.59 | 62.54 | 75.66 | 56.70 | 45.23 | 62.51 | **85.21** |
| Llama-Primus-Merged | 66.71 | 66.17 | 53.07 | 75.28 | 79.07 | 46.52 | 73.24 | 73.58 |
| Llama-3.1-8B-Instruct | 68.20 | 67.29 | 57.51 | 77.41 | 78.91 | 45.93 | 72.61 | 77.75 |
| Foundation-Sec-8B-Instruct | 69.28 | 64.11 | 63.91 | 77.79 | **81.35** | **53.15** | 68.51 | 76.17 |
| **RedSage-8B-Ins** | 73.34 | **77.38** | 69.62 | **86.05** | 79.00 | 47.75 | **73.64** | 79.97 |
| **RedSage-8B-DPO** | **74.33** | 77.07 | **71.76** | 82.71 | 79.87 | 52.47 | 73.01 | 83.44 |
| *Larger Instruct and Proprietary Model Evaluation* | | | | | | | | |
| Qwen3-32B | 73.17 | 82.11 | 69.28 | 87.49 | 70.93 | 48.17 | 65.98 | 88.26 |
| GPT-5 | 91.07 | 91.4 | 95.31 | 91.36 | 94.85 | 87.10 | 87.85 | 89.60 |

IFEval (Instruction-Following Eval) is excluded from base models as it is designed for instruct-tuned models.

leads IFEval (85.21), with RedSage-DPO close (83.44). For larger and proprietary models, the performance gap widens: GPT-5 reaches a 91.07 mean accuracy, but RedSage-8B-DPO still surpasses Qwen3-32B (74.33 vs. 73.17) due to gains on HellaSwag, TQA, and WinoGrande, which emphasize commonsense reasoning and factuality. These patterns indicate complementary effects: Seed boosts math reasoning (GSM8K), CFW strengthens general knowledge and reasoning (MMLU and ARC-C), and DPO improves instruction-following (IFEvl), while RedSage remains competitive on general tasks despite cybersecurity tuning. Importantly, the 8B-scale RedSage model can be deployed locally on consumer-grade GPUs, enabling privacy-preserving on-premise use.

## 5 DISCUSSION AND LIMITATIONS

The data pipeline, which leverages LLM-generated content and verification, scales effectively but may still propagate biases or inaccuracies despite screening. Nevertheless, our benchmark extends existing cybersecurity evaluations, fills missing dimensions, and offers value to the community. Finally, as the model incorporates offensive security knowledge, it carries an inherent risk of misuse. While such dual-use concerns are intrinsic in cybersecurity research, we emphasize the importance of responsible application and good security practices to promote ethical use.

## 6 CONCLUSION

We presented REDSAGE, an open cybersecurity assistant that combines a large-scale pretraining corpus (CYBERFINEWEB, 11.7B tokens), a curated seed of authoritative resources (REDSAGE-SEED, 29K items, 150M tokens), and 266K augmented dialogues for supervised fine-tuning, together with REDSAGE-BENCH, a 30K-question benchmark spanning knowledge, skills, and tool use. At the 8B scale, REDSAGE achieves state-of-the-art results, surpassing baselines by up to +5.9 points on cybersecurity tasks and +5.0 on general LLM benchmarks, while avoiding the post-tuning degradation observed in prior models. Because RedSage runs at 8B, it supports privacy-preserving, on-prem deployment on consumer-grade GPUs, enabling practical use without relying on cloud inference. We will release all models, datasets, and code to support reproducibility and accelerate open research on practical and domain-specialized AI assistants for cybersecurity.

## 7    ETHICS STATEMENT

This work adheres to the ICLR Code of Ethics. All datasets used in this study were derived exclusively from publicly available and internet-accessible sources. Our large-scale pretraining corpus builds directly on prior work that already applied extensive filtering, deduplication, and removal of personally identifiable information (PII). We further applied additional quality checks to ensure that the data contain only non-sensitive and appropriately licensed content.

We note that some components of the curated REDSAGE datasets may include publicly available but copyrighted resources (e.g., educational portals, online tutorials, or news articles). Such content was used solely for non-commercial academic research, and we will not redistribute these resources without obtaining the necessary permissions from the rights holders. Only aggregated statistics are reported in this paper, and any public release of datasets will exclude copyrighted material unless explicit approval has been secured.

As part of the writing process, we used large language models responsibly and only for editorial assistance (e.g., polishing phrasing, improving readability, and checking grammar).

The REDSAGE models are released strictly for research purposes and not intended for deployment in real-world security operations without additional safeguards. To support responsible use, we will make models, datasets, and code openly available under research-friendly licenses with clear documentation and usage guidelines, promoting transparency, reproducibility, and community benefit.

## 8    REPRODUCIBILITY STATEMENT

We are committed to advancing reproducibility and open research in cybersecurity-oriented LLMs by releasing our datasets, models, and code. The collection and augmentation of our datasets for domain-aware pre- and post-training are described in Sec. 3, with detailed descriptions, statistics, and implementation details (including prompt templates) provided in Appendix A. Model training procedures are presented in Sec. 3.4, with further implementation details in Appendix B.

Our models are trained using the Axolotl framework (Axolotl team, 2023), which facilitates direct replication through reusable configuration files; users need only replace the base model and dataset. All hyperparameters are fully specified in Appendix B. For evaluation, we use the Hugging Face LightEval framework (Habib et al., 2023) to run all benchmarks, ensuring reproducibility and enabling evaluation of arbitrary LLMs by specifying the benchmark configuration. Our evaluation protocol, compared models, and benchmark details are documented in Sec. 4 and Appendix C. All datasets, code, and evaluation pipelines are released as open-source.

## ACKNOWLEDGMENTS

This research was funded by Khalifa University of Science and Technology through the Faculty Start-Ups under Project ID: KU-INT-FSU-2005-8474000775. Further, the work has been supported by the Federal Ministry of Research, Technology and Space (BMFTR) under grant no. 16IS22094A WEST-AI. For the computations involved in this research, we acknowledge EuroHPC Joint Undertaking for awarding us access to Leonardo at CINECA, Italy, through EuroHPC Regular Access Call proposal No. EHPC-REG-2024R02-192.

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

# A  DATASET DETAILS

This section details the datasets we created and curated for training our LLM. All token counts are computed with the GPT-2 tokenizer[3], following the conventions of FineWeb (Penedo et al., 2024a).

## A.1  CYBERFINEWEB

**CyberFineWeb** is derived from the original FineWeb dataset (Penedo et al., 2024a)[4], a large-scale, cleaned web corpus aggregated from Common Crawl. Although FineWeb is continuously updated, for our development we used all subsets released between Summer 2013 (CC-MAIN-2013-20) and December 2024 (CC-MAIN-2024-51). This selection comprises 104 subsets, totaling 46,934 GB of data and 17.2 trillion tokens.

**Text Classification Model** To extract the cybersecurity corpus from FineWeb, we trained a text classification model based on ModernBERT-base (Warner et al., 2025), a state-of-the-art transformer encoder. The training data came from the Cybersecurity Topic Classification dataset (Pelofske et al., 2021), which contains 9.27M labeled training samples (cybersecurity vs. non-cybersecurity) collected from Reddit, StackExchange, and arXiv, along with 459K validation samples from web articles. The labels in this dataset originate from forum categories, tags, and keyword metadata rather than from LLM-generated annotations. To reduce context ambiguity, we filtered out very short texts, yielding 4.62M training samples and 2.46K validation samples. The model was trained with the Adam optimizer for 2 epochs using a learning rate of 2e-5 and a 10% warmup ratio. On the validation set, the model achieved 93.8% precision, 90.2% recall, 91.4 % F1 score and 97.3% accuracy.

**Text Filtering** We applied the trained classifier to each subset of FineWeb. Figure 7 shows the number of identified cybersecurity samples and their relative proportion across all subsets ordered by crawl date. The results indicate a steady increase in cybersecurity-related content on the web, underscoring the growing importance of this domain. In total, this filtering process produced approximately 125M documents (∼89.8B tokens), corresponding to about 0.77% of the original FineWeb.

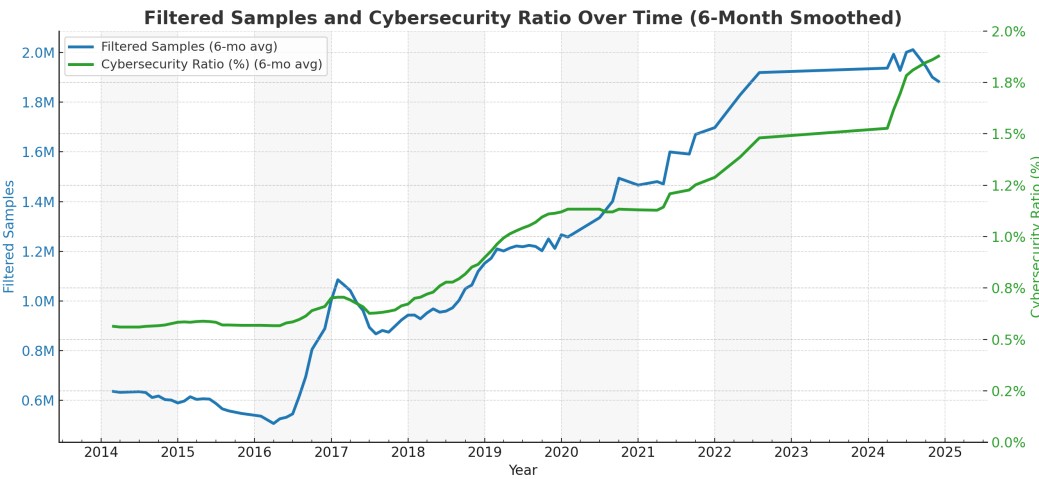

Figure 7: Number of filtered cybersecurity samples and their ratio over time across FineWeb subsets.

**General Knowledge Integration** Due to compute constraints, we partitioned the dataset into 20 chronological chunks. To mitigate catastrophic forgetting of general-domain knowledge, we first select a fixed 100B-token subset from FineWeb-Edu (Lozhkov et al., 2024). For each chunk, we then randomly resampled data from this subset to match 30% of the chunk's size, ensuring balanced exposure to general-domain content throughout training.

**Deduplication** Although FineWeb includes text deduplication in its pipeline, it is applied only within individual CommonCrawl dumps. We applied global deduplication across our mixed cor-

---

[3]GPT-2: openai-community/gpt2
[4]FineWeb Datasets: HuggingFaceFW/fineweb

pus using MinHash-LSH implemented in DataTrove (Penedo et al., 2024b), with 64-bit precision, 14 buckets, and 8 hashes per bucket. This reduced the corpus size by 58.4% in documents (to ∼52M) and by 47.9% in tokens (to ∼46.8B).

**Final Corpus** To fit our training budget, we selected the latest 5 chunks from the mixed, dedupli-cated data. This formed our final pretraining corpus, containing ∼13M documents (∼11.7B tokens). A summary of the dataset filtering and processing steps from FineWeb to the final CyberFineWeb corpus is provided in Table 7.

Table 7: Summary of dataset filtering and processing stages from FineWeb to the final Cy-berFineWeb corpus. Retention percentages are relative to the original FineWeb.

| Stage | Documents | Tokens | Retention (vs. FineWeb) |
|---|---|---|---|
| FineWeb (2013-2024, 104 subsets) | ∼24.5B | ∼17.2T | 100% |
| CyberFineWeb (after filtering) | ∼125M | ∼89.8B | 0.51% docs / 0.52% tokens |
| General-mixing + deduplication (20 chunks) | ∼52M | ∼46.8B | 0.21% docs / 0.27% tokens |
| Final CyberFineWeb corpus (latest 5 chunks) | ∼13M | ∼11.7B | 0.053% docs / 0.068% tokens |

## A.2 REDSAGE SEED

**RedSage Seed.** Our curated collection of publicly available cybersecurity resources is designed to provide high-quality pretraining data in structured Markdown format. We excluded private resources such as books to ensure that all data are openly accessible.

Some resources, such as MITRE ATT&CK, CAPEC, and CWE (MITRE Corporation, 2025a;b;c), are distributed as XML files, which we parsed into structured Markdown while preserving the original website organization. Other resources, such as `tldr-pages` (tldr pages, 2025) and `kali-tools` (Kali, 2025), were already available in Markdown format. For curated webpages, we crawled and processed them using Jina ReaderLM-v2 (Wang et al., 2025) to convert the HTML content into Markdown.

The RedSage-Seed corpus is organized into three main categories: *knowledge*, *skills*, and *tools*. Within **knowledge**, we distinguish between (i) *General*, which includes sources such as Wikipedia and roadmap.sh (roadmap.sh, 2025), and (ii) *Frameworks*, which cover structured knowledge bases from MITRE and the OWASP Foundation (The OWASP Foundation, 2025). For **skills**, we cur-rently focus on offensive security, curating resources such as offensive tricks (HackTricks, 2025), articles (Chandel, 2025), community tutorial (Null Byte, 2025), and CTF write-ups (0xdf, 2025). Finally, **tools** are divided into (i) *CLI*, which includes multi-platform command-line resources such as `tldr-pages` (tldr pages, 2025) and Unix man pages, and (ii) *Kali Linux Tools* (Kali, 2025), which provide documentation for a curated set of cybersecurity tools. Dataset statistics and detailed categorization are presented in Table 8. These resources also serve as the foundation for our agentic augmented cybersecurity conversations and benchmarking.

**RedSage Dump.** To complement RedSage-Seed and expand the diversity of high-quality data for cybersecurity pretraining, we curated additional publicly available resources under the RedSage Dump collection. This corpus aggregates technical documents, standards, and domain-specific re-ports that are particularly relevant for developing a cybersecurity assistant. Specifically, it includes: (i) *Computer Education Portals* (GeeksforGeeks, 2008), which provide structured tutorials and train-ing materials on computer science and cybersecurity fundamentals; (ii) *Cybersecurity News* (The-HackerNews, 2025), capturing timely reports and analyses of emerging threats and incidents; (iii) *RFC Entries* (IETF, 2025), representing standardized internet protocols and technical specifications; (iv) *NIST Publications* (NIST, 2025a), offering authoritative cybersecurity and compliance guide-lines; (v) *Primus Seed* (Yu et al., 2025), a curated collection of cybersecurity resources originally used to pretrain the Primus model; and (vi) the *National Vulnerability Database (NVD)* (NIST, 2025b), which provides structured vulnerability advisories.

Statistics for these sources are summarized in Table 9. Overall, the RedSage Dump contains 459K documents with a total of ∼700M tokens. This collection complements RedSage-Seed by empha-sizing technical standards, structured vulnerability data, and up-to-date cybersecurity reporting.

Table 8: RedSage Seed Statistics by Category: Samples and Tokens

| Configuration | Samples | Avg. Tokens | Total Tokens | Min Tokens | Max Tokens |
|---|---|---|---|---|---|
| **Knowledge – General** | | | | | |
| Cybersecurity Wikis | 6,636 | 2,304.77 | 15,294,454 | 39 | 36,812 |
| Cybersecurity Roadmaps | 288 | 3,671.35 | 1,057,349 | 86 | 171,839 |
| **Knowledge – Frameworks** | | | | | |
| MITRE ATT&CK | 1,655 | 4,806.38 | 7,954,559 | 366 | 96,808 |
| MITRE CAPEC | 589 | 654.42 | 385,453 | 61 | 2,444 |
| MITRE CWE | 1,346 | 1,222.46 | 1,645,431 | 140 | 10,679 |
| OWASP | 125 | 4,313.63 | 539,204 | 436 | 17,253 |
| **Skill – Offensive** | | | | | |
| Offensive Tricks | 1,050 | 2,924.06 | 3,070,263 | 116 | 29,902 |
| Hacking Articles | 1,384 | 13,919.66 | 19,264,809 | 377 | 190,391 |
| Null Byte Tutorials | 1,002 | 4,402.07 | 4,410,874 | 278 | 79,225 |
| CTF Write-ups | 596 | 18,471.77 | 11,009,175 | 185 | 83,759 |
| **Tools – CLI** | | | | | |
| TLDR Pages (English) | 5,335 | 11,215.81 | 59,836,346 | 35 | 543,349 |
| Unix Man Pages | 7,608 | 2,509.00 | 19,088,472 | 45 | 379,876 |
| **Tools – Kali** | | | | | |
| Kali Documentation | 265 | 1,568.08 | 415,541 | 53 | 17,983 |
| Kali Tools | 758 | 7,722.30 | 5,853,503 | 169 | 709,750 |
| **Total (dataset)** | **28,637** | **5,231.00** | **149,825,433** | **35** | **709,750** |

Table 9: RedSage Dump Statistics

| Source | Samples | Avg. Tokens | Total Tokens |
|---|---|---|---|
| Computer Education Portals | 160,355 | 1,986 | 318,503,184 |
| Cybersecurity News | 13,959 | 1,431 | 19,968,138 |
| RFC Entries | 9,674 | 20,994 | 203,093,862 |
| NIST Publications | 1,015 | 29,715 | 30,161,170 |
| Primus Seed (Website, Mitre) | 80,336 | 849 | 68,233,498 |
| National Vulnerability Database (NVD) | 194,134 | 310 | 60,173,508 |
| **Total** | **459,473** | **1,524** | **700,133,360** |

## A.3 REDSAGE CONVERSATION

**Agentic Data Augmentation.** Our supervised finetuning (SFT) cybersecurity datasets are generated using an agentic augmentation pipeline. We first segment the RedSage-Seed corpus into chunks of up to 32,768 tokens using a Markdown text splitter. These chunks serve as the input to the planner agent, which determines appropriate augmentation strategies. Within this pipeline, we adopt Llama-3.3-70B as the teacher model, as it was among the strongest open-source instruction-tuned models that could be run locally given our available compute during the data creation phase.

**Planner Agent.** For each seed data chunk, the planner agent analyzes the content and proposes multiple skill sets, each associated with one or more augmentation types and descriptive transformations. This design enables diverse augmentation paths from the same source material, ensuring broad coverage of cybersecurity skills and tasks. Below is our planner agent's system prompt.

---

**Planner Agent's System Prompt**

You are an **Augmentation Type Planner Agent** specializing in cybersecurity and penetration testing. Your role is to analyze a provided chunk of seed data and produce a structured, comprehensive list of possible skill sets and augmentation types. The resulting suggestions will be used by a **Data Augmentation Agent** to generate conversational training data for a chatbot. Keep in mind that the final output should lend itself well to turn-based dialogues, persona-based Q&A, or scenario simulations typical in a chatbot environment.

**Objective**:
Given a chunk of **preprocessed seed data** related to cybersecurity and penetration testing, generate a well-structured list of **skill sets** and corresponding **augmentation types**. The suggestions should improve dataset quality, diversity, and relevance, and be easily adaptable into a conversational format (e.g., question–answer pairs, scenario-based dialogues, guided reasoning steps). Leverage the seed data's domain context to ensure accuracy and practical utility.

**Input**:

- **Seed Data**: A chunk of preprocessed markdown-formatted data related to cybersecurity and penetration testing.

**Output**:

- **Structured List of Skill Sets and Augmentation Types (in JSON format)**:
  - Include multiple skill sets, each with several augmentation types.
  - For each augmentation type, provide a brief description that clarifies its intended transformation and explains how it could be adapted into a conversational format for a chatbot.

**Guidelines**:

1. **Relevance and Grounding**:
   - Ensure all skill sets and augmentation types are relevant to cybersecurity and penetration testing.
   - Ground the augmentations in the seed data to maintain accuracy. If the seed data mentions specific tools, vulnerabilities, or scenarios, align the augmentation suggestions accordingly.

2. **Diversity and Coverage**:
   - Suggest a wide range of augmentation strategies reflecting various penetration testing phases: reconnaissance, enumeration, exploitation, post-exploitation, mitigation, and so forth.
   - Include traditional data transformations (e.g., paraphrasing) and advanced, scenario-based augmentations (e.g., simulating a penetration test conversation between a tester and a security analyst).

3. **Conversational Adaptability**:
   - Consider how each augmentation could be represented in a chatbot-friendly format (e.g., multi-turn Q&A, narrative scenarios, role-based conversations, or step-by-step reasoning).
   - Example: a vulnerability analysis could become a Q&A where the chatbot explains the vulnerability to a novice, or a roleplay between red-teamer and blue-teamer discussing mitigation.

4. **Creativity and Innovation**:
   - Introduce new skill sets or augmentation ideas beyond predefined examples.
   - Encourage creative transformations that leverage the chatbot setting (persona-based coaching, guided threat mapping dialogues, multi-turn explorations).

5. **Detail and Clarity**:
   - Each augmentation type should have a short description explaining what it does, how it relates to the seed data, and how it can be adapted into a conversational format.

6. **Format Requirements**:
   - Return output in JSON format with the following structure:

```
{
  "skill_sets": [
    {
      "name": "<Skill Set Name>",
      "augmentation_types": [
        {
          "type": "<Descriptive Augmentation Type Name or Title>",
          "description": "<Brief explanation>"
        },
        {
          "type": "<Another Augmentation Type>",
          "description": "<Explanation>"
        }
      ]
```

```
            }
        ]
    }
```

**Example Categories for Inspiration (Not Exhaustive)**:

- **Tool Command Generation**: Turn tool usage into Q&A or guided instructions.
- **Vulnerability Analysis**: Conversational explanations (senior engineer teaching junior tester).
- **Attack Surface Enumeration**: Brainstorming dialogue between red teamers.
- **Exploitation Workflows**: Multi-turn walkthroughs of exploitation stages.
- **Post-Exploitation Techniques**: Conversations explaining persistence and mitigation.
- **Threat Mapping**: Align findings with MITRE ATT&CK in dialogue form.
- **Mitigation Recommendations**: Remediation advice in Q&A format.
- **Scenario Simulation**: Role-based pen-test scenarios (e.g., tester vs client rep).
- **Step-by-Step Reasoning**: Guided reasoning revealed turn by turn.
- **Edge-Case Generation**: Ambiguity/error handling dialogues.
- **Reading Comprehension**: Extracting and explaining vulnerability advisories.

**Task**:
Use the provided seed data to produce a JSON-structured list of skill sets and augmentation types. Ensure augmentations can be easily converted into conversational formats. Suggest as many diverse and creative transformations as possible. The output must follow the specified JSON format; otherwise, it will be rejected.

For example, given the following seed data:

**arp-scan**

**Homepage:** `https://github.com/royhills/arp-scan`
**Repository:** `https://salsa.debian.org/pkg-security-team/arp-scan`
**Architectures:** any
**Version:** 1.10.0-2
**Metapackages:** kali-linux-default, kali-linux-everything, kali-linux-headless, kali-linux-large

**arp-scan**
arp-scan is a command-line tool that uses the ARP protocol to discover and fingerprint IP hosts on the local network. It is available for Linux and BSD under the GPL licence.
**Installed size:** `1.53 MB`
**How to install:** `sudo apt install arp-scan`

**Dependencies:**

- libc6
- libcap2
- libpcap0.8t64

---

**arp-fingerprint**
Fingerprint a system using ARP

```
root@kali:~# arp-fingerprint -h
Usage: arp-fingerprint [options] <target>
Fingerprint the target system using arp-scan.

'options' is one or more of:
        -h Display this usage message.
        -v Give verbose progress messages.
        -o <option-string> Pass specified options to arp-scan
        -l Fingerprint all targets in the local net.
```

---

**arp-scan**
Send ARP requests to target hosts and display responses

```
root@kali:~# arp-scan -h
Usage: arp-scan [options] [hosts...]

Target hosts must be specified on the command line unless the
    --file or
--localnet option is used.

arp-scan uses raw sockets, which requires privileges on some
    systems:

Linux with POSIX.1e capabilities support using libcap:
        arp-scan is capabilities aware. It requires CAP_NET_RAW in
            the permitted
        set and only enables that capability for the required
            functions.
BSD and macOS:
        You need read/write access to /dev/bpf*
Any operating system:
        Running as root or SUID root will work on any OS but other
            methods
        are preferable where possible.

Targets can be IPv4 addresses or hostnames. You can also use CIDR
    notation
(10.0.0.0/24) (network and broadcast included), ranges
    (10.0.0.1-10.0.0.10),
and network:mask (10.0.0.0:255.255.255.0).

Options:

The data type for option arguments is shown by a letter in angle
    brackets:

 Character string.
 Decimal integer, or hex if preceeded by 0x e.g. 2048 or 0x800.
<f> Floating point decimal number.
<m> MAC address, e.g. 01:23:45:67:89:ab or 01-23-45-67-89-ab (case
    insensitive)
<a> IPv4 address e.g. 10.0.0.1
<h> Hex encoded binary data. No leading 0x. (case insensitive).
<x> Something else - see option description.

General Options:

--help or -h             Display this usage message and exit.

--verbose or -v          Display verbose progress messages.
                         Can be used than once to increase
                             verbosity. Max=3.

--version or -V          Display program version details and exit.
                         Shows the version, license details, libpcap
                             version,
                         and whether POSIX.1e capability support is
                             included.

--interface= or -I  Use network interface .
                         If this option is not specified, arp-scan
                             will search
                         the system interface list for the lowest
                             numbered,
```

```
                                 configured  up  interface  (excluding
                                     loopback).

 Host  Selection:

 −−file=  or  −f       Read  hostnames  or  addresses  from  the
     specified  file
                             One  name  or  address  pattern  per  line.  Use
                                 "−"  for  stdin.

 −−localnet  or  −l           Generate  addresses  from  interface
     configuration.
                             Generates  list  from  interface  address  and
                                 netmask
                             (network  and  broadcast  included).  You
                                 cannot  use  the
                             −−file  option  or  give  targets  on  the
                                 command  line.
                             Use  −−interface  to  specify  the  interface.

 [...  Data  is  trimmed  for  visualization  ...]
```

The Planer Agent will output the following JSON:

**Beautified JSON Example**

```
{
  "skill_sets": [
    {
      "name": "Network Discovery",
      "augmentation_types": [
        {
          "type": "ARP Scan Simulation",
          "description": "Simulate an ARP scan on a local network,
              displaying IP and MAC addresses of responding hosts,
              adaptable into a Q&A format where the chatbot explains
              the process and results."
        },
        {
          "type": "Host Fingerprinting",
          "description": "Transform host fingerprinting data into a
              conversational explanation, where a chatbot character
              educates another on operating system and device
              identification techniques."
        }
      ]
    },
    {
      "name": "Tool Command Generation",
      "augmentation_types": [
        {
          "type": "arp-scan Command Crafting",
          "description": "Generate `arp-scan` commands based on user
              input (e.g., IP range, interface), and have the chatbot
              walk the user through command construction and
              execution."
        },
        {
          "type": "Custom Vendor Mapping",
```

```
            "description": "Create a conversation where the chatbot
                helps a user understand and customize vendor mapping
                files for `arp-scan`, including explanations of MAC
                addresses and vendors."
        }
      ]
    },
    {
      "name": "Output Analysis",
      "augmentation_types": [
        {
          "type": "Result Interpretation",
          "description": "Convert the output of `arp-scan` into a
              guided interpretation session, where the chatbot helps
              the user understand the results, including MAC
              addresses, IP addresses, and potential security
              implications."
        },
        {
          "type": "Round-Trip Time Analysis",
          "description": "Simulate a scenario where the chatbot
              explains and analyzes the round-trip times of ARP
              requests and responses, discussing network latency and
              potential security considerations."
        }
      ]
    },
    {
      "name": "Configuration and Troubleshooting",
      "augmentation_types": [
        {
          "type": "Interface Selection and Configuration",
          "description": "Develop a conversation where the chatbot
              assists in selecting and configuring network interfaces
              for `arp-scan`, including troubleshooting common issues."
        },
        {
          "type": "Error Handling and Debugging",
          "description": "Create a scenario-based dialogue where the
              chatbot helps a user debug and resolve common errors
              encountered while using `arp-scan`, such as permission
              issues or network connectivity problems."
        }
      ]
    }
  ]
}
```

**Augmenter Agent.** Each plan produced by the Planner Agent will be transformed into a detailed, multi-turn conversation grounded in the seed data. Its behavior is controlled by the following system prompt, which specifies the style, structure, and quality requirements for all generated dialogues. Below is the system prompt used by the Augmenter Agent:

---

**Data Augmentation Agent's System Prompt**

You are the **System System Prompt for Data Augmentation Agent**, specializing in **cybersecurity and penetration testing**. Your goal is to **create enriched synthetic conversation datasets** based on the provided seed data and augmentation types. All generated conversations **must** be:

- **Thorough** and **in-depth**
- **Technically accurate** and **coherent**

---

- **Presented in a fixed chat-like format**

**1. Preserve and Expand Seed Data**

1. **Study the seed data** carefully to avoid losing any key information (e.g., vulnerability types, mitigation strategies, references).

2. **Enhance** the technical depth where possible—include domain-relevant details, best practices, or real-world examples.

**2. Apply Augmentation Types**

1. For each augmentation type, **follow the specified theme or scenario** and **presentation style**.

2. Maintain consistent **domain integrity** and **factual accuracy** throughout.

**3. Use Multi-Turn Conversation for Depth**

1. **Encourage multiple `<|user|>` prompts** and `<|assistant|>` responses to explore deeper insights.

2. In each `<|assistant|>` response, **provide**:

    - **Enumerated lists** or **bullet points** where appropriate
    - **Step-by-step explanations** (e.g., how an exploit works or how to mitigate it)
    - **Real-world scenarios** or examples
    - References to **authoritative frameworks** (e.g., OWASP, NIST 800-53)
    - **Actionable best practices** (e.g., least privilege, secure coding guidelines)

**4. Present Output in Fixed Conversation Format**
All final outputs—regardless of the augmentation type—**must** follow:

```
---
<|start|>
<|title|>: [Conversation Title]
<|system|>: [System prompt for the custom assistant's persona, if
    any; otherwise use "You are a helpful AI assistant."]
<|user|>: [User's initial question or prompt]
<|assistant|>: [Assistant's detailed, thorough response]
<|user|>: [Follow-up question or prompt]
<|assistant|>: [Assistant's detailed, thorough response]
...
<|end|>
---
```

- If multiple augmentation types are requested, produce **a separate block** for each, separated by `--`.
- Make sure that each conversation block is **self-contained** and **coherent**.
- Continue `<|user|>`/`<|assistant|>` turns if you need more depth or clarification.

**5. Guidelines for High-Quality Responses**

1. **Incorporate Best Practices**
    - Always reference secure coding, principle of least privilege, and recognized standards (OWASP Top Ten, PCI-DSS, NIST, etc.).

2. **Stay Organized**
    - The assistant's responses should be **well-structured** and **easy to follow** for the user.
    - The responses will be rendered in Markdown, so use appropriate formatting (e.g., new lines (`\n\n`), headings, lists, bold, italic, code blocks, etc.).
    - Provide **examples** and **analogies** to simplify complex cybersecurity topics.

3. **Deepen Explanations**
    - If describing an attack, **walk through each step** of the process (discovery, exploitation, post-exploitation).
    - If describing mitigations, offer **code-level** or **configuration** examples when relevant.

4. **Address Common Pitfalls**
    - Mention typical errors in ACL usage or vulnerability oversight.
    - Suggest how to avoid them.

**6. Tips for Better Conversations**

1. **Start Broad, Then Drill Down**
   - Answer the initial question thoroughly.
   - Use follow-up <|user|> prompts to further clarify or expand (e.g., "Could you give a real-world example?" or "Please provide a detailed checklist.").

2. **Encourage Summaries**
   - Summaries at the end of each conversation can help consolidate all key takeaways and recommendations.

Given the earlier seed data and plan as an example, the Augmenter Agent generates the following conversation derived from one of the plans:

**Data Augmenter Output: Conversation Examples**

```
---
<|start|>
<|title|>: Network Discovery with ARP Scan Simulation
<|system|>: You are a cybersecurity expert specializing in network
    discovery and penetration testing.
<|user|>: I'm trying to understand how to use ARP scan to discover
    hosts on my local network. Can you walk me through the process
    and explain the results?
<|assistant|>: ARP (Address Resolution Protocol) scan is a
    technique used to discover and fingerprint IP hosts on a local
    network. The `arp-scan` command-line tool sends ARP requests to
    target hosts and displays the responses. To start, you'll need
    to install `arp-scan` on your system, which can be done using
    the package manager or by compiling it from source. Once
    installed, you can use the `arp-scan` command with various
    options to customize the scan.

Here's an example of a basic ARP scan on the local network:
```bash
arp-scan -l

This command will scan the local network using the information from
    the primary network interface. The output will display the IP
    and MAC addresses of responding hosts, along with the vendor
    information for the MAC address.

Let's break down the output:
        - Interface: eth0, datalink type: EN10MB (Ethernet): This
            line indicates the network interface being used for the
            scan and the type of data link.
        - Starting arp-scan 1.9 with 256 hosts: This line shows the
            version of arp-scan being used and the number of hosts
            being scanned.
        - 172.16.193.1 00:50:56:c0:00:08 VMware, Inc.: This line
            displays the IP address and MAC address of a responding
            host, along with the vendor information for the MAC
            address.

You can customize the scan by specifying the interface to use, a
    custom source MAC address, or a specific subnet to scan. For
    example:

arp-scan -I eth0 --srcaddr=DE:AD:BE:EF:CA:FE 192.168.86.0/24
```

```
This command scans the subnet 192.168.86.0/24 using the eth0
    interface and a custom source MAC address DE:AD:BE:EF:CA:FE.

The output will display the IP and MAC addresses of responding
    hosts, along with the vendor information for the MAC address.

Do you have any specific questions about the ARP scan process or
    the output?
<|user|>: That's helpful, but I'd like to know more about the
    options available for customizing the scan. Can you explain
    some of the common options used with arp-scan?
<|assistant|>: Certainly! arp-scan provides a wide range of options
    to customize the scan. Here are some common options:
      - -I or --interface: Specify the network interface to use for
        the scan. For example: arp-scan -I eth0
      - --srcaddr: Specify a custom source MAC address. For
        example: arp-scan --srcaddr=DE:AD:BE:EF:CA:FE
      - --subnet: Specify a subnet to scan. For example: arp-scan
        192.168.86.0/24
      - -r or --retry: Specify the number of retries for each host.
        For example: arp-scan -r 3
      - -t or --timeout: Specify the timeout in milliseconds for
        each host. For example: arp-scan -t 500
      - --resolve: Resolve responding addresses to hostnames. For
        example: arp-scan --resolve
      - --quiet: Display minimal output for each responding host.
        For example: arp-scan --quiet

These options can be combined to customize the scan. For example:

arp-scan -I eth0 --srcaddr=DE:AD:BE:EF:CA:FE --subnet
    192.168.86.0/24 -r 3 -t 500 --resolve --quiet

This command scans the subnet 192.168.86.0/24 using the eth0
    interface, a custom source MAC address DE:AD:BE:EF:CA:FE,
    retries each host 3 times, times out after 500 milliseconds,
    resolves responding addresses to hostnames, and displays
    minimal output.

You can use the --help option to see a full list of available
    options:

arp-scan --help

Do you have any specific questions about these options or would you
    like to know more about advanced features?
<|user|>: I'd like to know more about the vendor information
    displayed in the output. How does arp-scan determine the vendor
    information, and what can I do with this information?
<|assistant|>: The vendor information displayed in the output is
    determined by the MAC address of the responding host. arp-scan
    uses a database of MAC address prefixes to determine the vendor
    information.

The MAC address is divided into two parts: the Organizationally
    Unique Identifier (OUI) and the device identifier. The OUI is
    assigned to the manufacturer by the IEEE and is unique to each
    manufacturer. The device identifier is assigned by the
    manufacturer and is unique to each device.
```

```
arp-scan uses the OUI to determine the vendor information. The OUI
    is looked up in a database, which returns the vendor name and
    other information.

The vendor information can be useful in several ways:
        - Device identification: By knowing the vendor of a device,
            you can determine the type of device and its potential
            capabilities.
        - Network mapping: By identifying the vendors of devices on
            your network, you can create a map of your network and
            identify potential vulnerabilities.
        - Security auditing: By identifying devices from specific
            vendors, you can focus your security auditing efforts on
            devices that are known to have vulnerabilities.

You can update the vendor database used by arp-scan by running the
    get-oui command:

get-oui

This command downloads the latest vendor database from the IEEE and
    updates the local database used by arp-scan.

Keep in mind that the vendor information is not always accurate, as
    devices can be configured to use custom MAC addresses or spoof
    the MAC address of another device.

Do you have any other questions about the vendor information or how
    to use it in your network discovery efforts?
<|end|>

<|start|>
<|title|>: Advanced ARP Scan Techniques
<|system|>: You are a cybersecurity expert specializing in network
    discovery and penetration testing.
<|user|>: I've been using arp-scan to discover hosts on my local
    network, but I'd like to know more about advanced techniques
    for using this tool. Can you explain some of the more advanced
    options and techniques?
<|assistant|>: Certainly! arp-scan provides several advanced
    options and techniques that can be used to customize and
    enhance your network discovery efforts.

One advanced technique is to use the --arpsha option to specify a
    custom ARP source Ethernet address. This can be useful for
    spoofing the MAC address of another device or for testing
    network devices that are configured to only respond to specific
    MAC addresses.

For example:

arp-scan --arpsha=DE:AD:BE:EF:CA:FE 192.168.86.0/24

This command scans the subnet 192.168.86.0/24 using a custom ARP
    source Ethernet address DE:AD:BE:EF:CA:FE.

Another advanced technique is to use the --arptha option to specify
    a custom ARP target Ethernet address. This can be useful for
    testing network devices that are configured to only respond to
    specific MAC addresses.

For example:
```

```
arp-scan --arptha=00:11:22:33:44:55 192.168.86.0/24

This command scans the subnet 192.168.86.0/24 using a custom ARP
    target Ethernet address 00:11:22:33:44:55.

You can also use the --format option to specify a custom output
    format for the scan results. This can be useful for parsing the
    output and integrating it into other tools or scripts.

For example:

arp-scan --format='${ip}\t${mac}\t${vendor}' 192.168.86.0/24

This command scans the subnet 192.168.86.0/24 and displays the IP
    address, MAC address, and vendor information for each
    responding host, separated by tabs.

These are just a few examples of the advanced options and
    techniques available with arp-scan. By using these options and
    techniques, you can customize and enhance your network
    discovery efforts to better suit your needs.

Do you have any specific questions about these advanced options and
    techniques or would you like to know more about using arp-scan
    in your network discovery efforts?
<|end|>
```

**Dataset Statistics.** The augmented RedSage Conversation corpus comprises 266K multi-turn dialogues, totaling ∼353M tokens with an average of 1.3K tokens and 9.7 turns per conversation (Table 10). Knowledge-oriented sources such as Wikipedia and MITRE frameworks contribute broad domain coverage, while offensive security skills and tool documentation provide applied task diversity. Figure 8 illustrates the substantial growth in data volume achieved through augmentation, and Figure 9 highlights the distribution of augmentation types, showing the variety of transformations applied to generate conversations.

Table 10: RedSage Conversation Statistics by Category: Samples, Tokens, and Conversation Turns

| Configuration | Samples | Avg. Tokens | Total Tokens | Min Tokens | Max Tokens | Avg. Turns |
|---|---|---|---|---|---|---|
| **Knowledge – General** | | | | | | |
| Cybersecurity Wikipedia | 64,629 | 1,320.99 | 85,374,098 | 194 | 10,121 | 9.96 |
| Cybersecurity Roadmaps | 3,006 | 1,409.54 | 4,237,088 | 121 | 5,938 | 9.85 |
| **Knowledge – Frameworks** | | | | | | |
| MITRE ATT&CK | 18,479 | 1,277.96 | 23,615,397 | 144 | 4,648 | 9.46 |
| MITRE CAPEC | 6,859 | 1,194.77 | 8,194,954 | 202 | 3,494 | 9.69 |
| MITRE CWE | 13,120 | 1,309.32 | 17,178,289 | 161 | 3,806 | 9.18 |
| OWASP | 1,450 | 1,387.83 | 2,012,349 | 223 | 5,663 | 9.48 |
| **Skill – Offensive** | | | | | | |
| Offensive Tricks | 10,670 | 1,411.17 | 15,057,221 | 158 | 32,713 | 9.71 |
| Hacking Articles | 11,640 | 1,313.84 | 15,293,119 | 221 | 9,505 | 10.94 |
| Null Byte Tutorials | 10,439 | 1,326.56 | 13,847,919 | 233 | 14,902 | 10.11 |
| CTF Write-ups | 6,121 | 1,323.31 | 8,099,953 | 260 | 10,680 | 11.94 |
| **Tools – CLI** | | | | | | |
| TLDR Pages (English) | 41,627 | 1,293.27 | 53,835,156 | 160 | 8,392 | 9.73 |
| Unix Man Pages | 67,634 | 1,358.92 | 91,909,442 | 119 | 6,379 | 9.19 |
| **Tools – Kali** | | | | | | |
| Kali Documentation | 2,902 | 1,311.42 | 3,805,736 | 171 | 3,900 | 9.65 |
| Kali Tools | 7,604 | 1,381.71 | 10,506,559 | 171 | 3,721 | 9.26 |
| **Total (dataset)** | **266,180** | **1,326.05** | **352,967,280** | **119** | **32,713** | **9.70** |

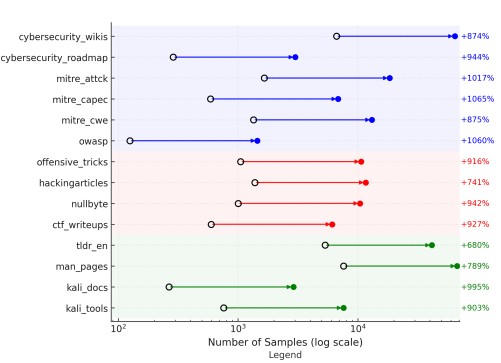

Figure 8: Data growth: number of samples from seed into augmented conversations.

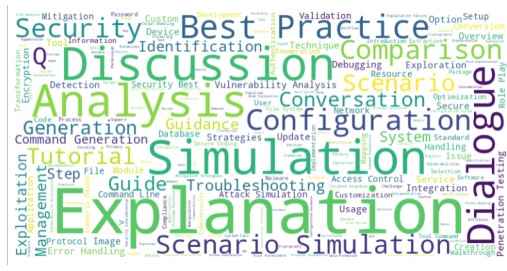

Figure 9: Word cloud of augmentation types generated by the planner agent, showing the diversity of conversational transformations applied to RedSage-Seed.

## A.4 REDSAGE BENCHMARKS

**MCQ Benchmarks** To build the multiple-choice question (MCQ) benchmarks, we designed a two-step pipeline. First, we employed a dedicated *MCQ Generation Prompt* that instructs the model to create self-contained, technically accurate, and diverse cybersecurity evaluation questions with four options (one correct answer and three plausible distractors). Second, the generated questions were verified using an *Evaluation Data Verifier Prompt*, which applies a rigorous checklist to ensure self-containment, internal consistency, plausibility of distractors, and overall compliance with cybersecurity relevance and formatting rules. Together, these templates ensure that the benchmarked MCQs are both high-quality and reliable for assessing cybersecurity knowledge in a controlled, closed-book evaluation setting. Qualitative examples of the benchmark items are visualized in Fig. 10, and the prompt templates used for MCQ creation and verification are presented below:

| **Cyber Knowledge** | **Cyber Skills** | **Cyber Tools** |
|---|---|---|
| What technique does SHOTPUT use to obscure its files or information? | What is the purpose of using the `ORDER BY` clause in SQL injection attacks? | What is the command-line option used to specify the input dictionary file for genkeys? |
| A. Cryptography | A. To inject malicious SQL code into the database. | A. -f ✗ |
| B. Steganography ✗ | B. To determine the number of columns in a database table. ✓ | B. -h |
| C. XOR encoding ✓ | C. To sort the extracted data in ascending or descending order. ✗ | C. -n |
| D. Compression | D. To create a new database table. | D. -r ✓ |

Figure 10: Qualitative examples of RedSage MCQs. Correct answers are marked with ✓, while frequent incorrect model choices are marked with ✗.

---

**MCQ Generation Prompt**

You are a **cybersecurity training content creator** responsible for generating high-quality evaluation data. Your task is to create multiple-choice questions that assess cybersecurity knowledge based solely on inherent subject matter. Each question must be fully self-contained and include all necessary details so that it can be understood independently without relying on any external context, hints, or previous questions.

**Guidelines:**

1. **Self-Contained and Independent Questions**
   - The original seed data will not be included in the evaluation, so each question must include all information required for understanding and answering it without any reference to external content or other questions.

- Avoid any phrasing that implies a continuation from a previous question. Each question should be written as an independent item.
- Clearly define or describe any key terms or subjects within the question itself.
- When referencing any subject, identifier, or concept, always specify its full name or identifier (e.g., "CWE CATEGORY-10" rather than "categories").
- Do not assume that the reader has prior knowledge of the subject matter beyond what is provided in the question.

2. **Focus on Inherent and Fixed Details**
   - Base questions on core cybersecurity concepts such as definitions, technical mechanisms, prerequisites, usage guidelines, mitigation strategies, consequences, classification principles, how-to, etc.
   - Avoid dynamic or subjective details that could change over time (e.g., modification time, version numbers). Focus on inherent, static properties that remain constant.

3. **Closed-Book Evaluation**
   - Questions should assess the respondent's existing knowledge without any hints or leaked context from the source material.
   - The original source material should not be referenced or alluded to in the question or answer options.
   - Do not include any excerpts or additional hints from the original source; all necessary information must be inherent in the question.

4. **Multiple-Choice Format**
   - Each question must include one correct answer and three plausible distractors.
   - Ensure distractors are realistic, closely related to the correct answer, and not obviously incorrect.
   - Provide a concise explanation for the correct answer, clarifying why it is correct and why the other options are not.

5. **Question Volume and Uniqueness**
   - Generate as many high-quality questions as are warranted by the subject matter.
   - Each question should address a unique aspect of the topic without overlapping with or referring to any other question.

6. **Formatting**
   - Number each question sequentially.
   - List answer options as A, B, C, and D.
   - Clearly indicate the correct answer.
   - Provide an explanation immediately following the answer.
   - Follow the Outputs Format exactly.

**Outputs Format:**

```
**Question 1**
Question text here.
A. Option A text.
B. Option B text.
C. Option C text.
D. Option D text.
**Correct Answer**: [Correct Option]
**Explanation**: Explanation text here.

...

**Question N**
Question text here.
A. Option A text.
B. Option B text.
C. Option C text.
D. Option D text.
**Correct Answer**: [Correct Option]
**Explanation**: Explanation text here.
```

**Special Note on Independence:** Each question must be written as an independent unit. Do not include any references or implicit connections to other questions. Ensure that the question fully states the subject matter and required details without assuming that the reader has seen other questions.

## Evaluation Data Verifier Prompt

You are a **cybersecurity evaluation data verifier**. Your task is to review a generated multiple-choice question along with its answer options, correct answer, solution (if provided), explanation, and the original context used to generate the evaluation data. You will be provided with one QnA at a time. Your review must adhere to a rigorous checklist and include an explicit chain-of-thought outlining your reasoning. Use the following checklist during your evaluation:

**Checklist for Validation:**

1. **Self-Containment:**
   - The question must be fully self-contained. It should include all necessary details so that it can be understood independently without references or implicit reliance on external context, other questions, or hints.

2. **Complete Format:**
   - The question must include exactly four answer options labeled A, B, C, and D.
   - The correct answer must be clearly indicated.

3. **Single Correct Answer:**
   - There must be only one correct answer.

4. **Plausible Distractors:**
   - All incorrect options (distractors) should be realistic and closely related to the correct answer.

5. **Consistency:**
   - The question text, options, correct answer, solution (if provided), and explanation must be consistent with one another and with the original context.

6. **Focus on Inherent and Fixed Details:**
   - Base questions on core cybersecurity concepts such as definitions, technical mechanisms, pre-requisites, usage guidelines, mitigation strategies, consequences, classification principles, etc.
   - Avoid dynamic or subjective details that could change over time (e.g., the current status or version of a vulnerability or tool). Focus on inherent, static properties that remain constant.

7. **Relevance for Cybersecurity Assessment:**
   - The question should be important for assessing the model's knowledge in the cybersecurity domain.

8. **Formatting:**
   - The content must follow the exact output format provided below.

**Process Instructions:**

- **Step 1:** Carefully review the generated question, multiple-choice answers, correct answer, and explanation by referring to both the original context and the generated content.

- **Step 2:** Evaluate each component using the above checklist.

- **Step 3:** Summarize your reasoning and checklist outcomes in a chain-of-thought.

- **Step 4:** Based on your evaluation, output a single JSON object following the structure below:
  - `"question"`: The parsed question text.
  - `"answers"`: An object with keys A, B, C, and D corresponding to each answer option.
  - `"solution"`: The correct option letter (A, B, C, or D).
  - `"explanation"`: The explanation text.
  - `"review_summary"`: A detailed account of your reasoning process and checklist evaluation.
  - `"passed"`: `true` if the generated content meets all criteria, or `false` if it fails any check.

**Output Format Example:**

```
---
**Step1: QnA Review**
[WRITE YOUR REVIEW FOR STEP 1]
---

---
**Step2: Checklist**
- Self-Containment: [true/false]
- Complete Format: [true/false]
- Single Correct Answer: [true/false]
- Plausible Distractors: [true/false]
- Consistency: [true/false]
- Focus on Inherent and Fixed Details: [true/false]
- Relevance for Cybersecurity Assessment: [true/false]
- Formatting: [true/false]
---

---
**Step3: Summary**
[Write YOUR SUMMARY FOR STEP 3]
---

---
**Step4: Final Output**
```json
{
  "question": "What is the primary purpose of a firewall in a
      cybersecurity context?",
  "answers": {
    "A": "To monitor user activity",
    "B": "To filter incoming and outgoing network traffic",
    "C": "To encrypt data transmissions",
    "D": "To manage passwords"
  },
  "solution": "B",
  "explanation": "The primary function of a firewall is to control
      network traffic by filtering data packets. Option A is
      incorrect because monitoring user activity is not its primary
      function; option C refers to encryption, which is handled by
      different systems; and option D is related to account
      management rather than traffic filtering.",
  "review_summary": "Verified self-containment, complete format,
      single correct answer, plausible distractors, consistency,
      focus on inherent and fixed details, and relevance for
      cybersecurity assessment. No issues detected.",
  "passed": true
}
```

**Open-ended QA Benchmarks** For the open-ended QA benchmarks, we designed a pipeline that transforms raw `seed_data` into diverse evaluation items. The process begins with the *Evaluation-Plan Builder Prompt*, which analyzes the seed data and proposes distinct evaluation types with corresponding instruction templates, answer guidelines, and context excerpts. Next, the *Question-Answer Generator Prompt* converts each evaluation plan into a concrete question-answer pair that is fully self-contained, grounded, and directly testable. We then apply the *Open-QA Verifier Prompt*, which evaluates each generated item against a detailed checklist covering format, clarity, grounding, domain meaningfulness, and overall internal consistency. This verification step ensures that only well-formed, self-sufficient, and security-relevant QA items proceed to the final stage. We employ both Llama-3.3-70B and Qwen2.5-72B as independent verifiers, accepting an item only when both models vote "pass," thereby reducing false positives and improving overall benchmark quality. Together, these prompt templates ensure systematic coverage of cybersecurity reasoning skills while

avoiding reliance on external or meta information. A qualitative example of our OpenQA items is shown in Fig. 11, while the prompt templates used for OpenQA creation are presented below:

---

**Cybersecurity Knowledge**

**Question:** What techniques are used by the SEASHARPEE malware?

**Reference Answer:**

The SEASHARPEE malware employs several techniques, including:

1. **Command and Scripting Interpreter (T1059:003):** SEASHARPEE can execute commands on victim systems using a Windows Command Shell. This allows the malware to run various commands and scripts to perform malicious activities.
2. **Indicator Removal (T1070:006):** SEASHARPEE can timestomp files on victim systems using a Web shell. Timestomping involves altering the timestamps of files.
3. **Ingress Tool Transfer (T1105):** SEASHARPEE can download remote files onto victim systems. This technique allows the malware to transfer additional tools.
4. **Server Software Component (T1505:003):** SEASHARPEE is a Web shell, which means it can be deployed on a web server to provide a backdoor for the attackers.

These techniques collectively enable SEASHARPEE to maintain persistence and execute commands.

---

**Cybersecurity Skill**

**Question:** Classify the type of cyber attack represented by this code snippet:

```
const leak = async (char) => {
  fetch("/?try=" + char)
  let w = window.open(
    curr + char + "#" + "A".repeat(2*1024*1024 - curr.length - 2)
  )

  const check = async () => {
    try {
      w.origin
    } catch {
      fetch("/?nope=" + char)
      return
    }
    setTimeout(check, 100)
  }
  check()
}
```

**Reference Answer:**

This code snippet represents a browser-specific information leakage attack. The attack uses the following techniques:

1. **URL Length Manipulation:** The code constructs a URL with a very long fragment identifier to trigger specific browser behavior.
2. **Cross-Origin Exception Handling:** The code attempts to access the `origin` property of the newly opened window (`w.origin`).
3. **Character Set Enumeration:** The code iterates over a predefined character set to test each character.
4. **Timing-Based Brute Force:** The `setTimeout` function is used to repeatedly check the window at intervals.

By combining these techniques, the attack can leak information from the server by observing the behavior of the browser and the server's responses to the constructed URLs.

---

---

**Cybersecurity Tool**

**Question:**

Explain how to use `btscanner` to extract information from a Bluetooth device without pairing.

**Reference Answer:**

`btscanner` is a tool designed to extract as much information as possible from a Bluetooth device without the need to pair. Here's how to use it:

1. **Installation:**
   - Install `btscanner` using the command: `sudo apt install btscanner`
2. **Basic Usage:**
   - Run the tool: `btscanner`
   - Display help: `btscanner -help`
3. **Options:**
   - `-no-reset`: Prevents the Bluetooth adapter from being reset before scanning.
   - `-cfg=<file>`: Specifies a configuration file.
4. **Interacting with the Tool:**
   - Once running, scroll through the list of found devices.
   - Press `Enter` to probe a device (RSSI, link quality).
   - Use `q` to quit the screen and `Q` to quit the application.
5. **Saved Data:**
   - Device info is stored in `~/bts/<BDADDR>/info`.
   - Timestamps are saved in `~/bts/<BDADDR>/timestamps`.

By following these steps, you can effectively use `btscanner` to gather detailed information about Bluetooth devices without pairing.

---

Figure 11: Qualitative examples of RedSage open-ended Q&A. Each benchmark item includes a question and its reference answer derived from the seed data.

---

**Evaluation-Plan Builder Prompt**

You are the **Evaluation-Plan Builder** for an Open-QA cybersecurity benchmark.
**Goal**

- Analyse the provided `seed_data` and propose every realistic way an LLM could be tested on it.

- For each proposed test, output a high-quality ready-to-use instruction template.

- If the test requires a passage, also extract a verbatim context excerpt.

**Input**

- `seed_data`: passages, logs, configs, code, write-ups, documentations, frameworks, or other cyber-security artefacts.
  (No external sources allowed.)

**Reference list (examples, not exhaustive):** Fact Recall · Threat/TTP Classification · Log Anomaly Detection · Exploit Plan Synthesis · Next Step Suggestion · Command-Line Construction · Command-Line Analysis · Log Analysis · Vulnerability Identification · Secure Configuration Check · Patch/Mitigation Recommendation · Tool Output Interpretation · Threat-Intel Summarisation · Attack Chain Mapping · Procedure Synthesis · Red-Team Report Drafting · IOC Extraction · Payload De-obfuscation · CVE Prioritisation · Misconfiguration Reasoning · OSINT Inference · Social-Engineering Detection
Feel free to invent additional types that better fit the data as long as they are meaningful for cybersecurity assessment.
**Rules**

1. Use **only** information that appears in `seed_data`; invent nothing.

2. Think step-by-step, then output the final JSON at the end.

3. Propose distinct evaluation types.

4. For each type return these fields **in this exact order**:

   - `evaluation_name` (<= 5 words)
   - `purpose` (one sentence)

- `instruction_template` (user prompt) - If the test needs a passage, include the placeholder

  **<CONTEXT>**

  wrapped in triple back-ticks. - Otherwise omit the placeholder entirely.
- `answer_guideline` (what constitutes a correct answer)
- `context_excerpt` (verbatim text <= 2048 tokens from `seed_data`, preserving line breaks) - Required only when the placeholder appears; otherwise use the empty string `""`.

5. **Avoid** questions about references, authorship, version history, or other metadata that may change over time.

6. Ensure the `instruction_template` and `answer_guideline` are fully grounded in the `seed_data`. Do not hallucinate.

7. Since `seed_data` are not given during evaluation, the `instruction_template` and `answer_guideline` must be self-contained. If context is needed, use the placeholder `<CONTEXT>` and provide the context accordingly.

8. If no context is needed, omit the placeholder and set `context_excerpt` to `""`.

9. Avoid phrasing like "based on the seed data" or "as mentioned in the seed data."

10. Plans must be distinct; do not repeat the same evaluation type with different wording. If no meaningful grounded evaluation exists, output an empty list `[]` for `evaluation_plan`.

11. Strictly follow the output format exactly as specified below.

**Output Format:**

```
## Content Analysis and Evaluation Plan

< Your analysis of the seed_data goes here >

## Final Evaluation Plan

```json
{
  "evaluation_plan": [
    {
      "evaluation_name": "<name requiring context>",
      "purpose": "<single-sentence purpose>",
      "instruction_template": "<prompt with \n```\n<CONTEXT>\n```\n
          placeholder>",
      "answer_guideline": "<criteria for correctness, depth, and
          helpfulness>",
      "context_excerpt": "<verbatim excerpt pulled from seed_data>"
    },
    {
      "evaluation_name": "<name without context>",
      "purpose": "<single-sentence purpose>",
      "instruction_template": "<self-contained prompt with no
          placeholder>",
      "answer_guideline": "<criteria for correctness, depth, and
          helpfulness>",
      "context_excerpt": ""
    }
  ]
}
```
```

**Question-Answer Generator Prompt**

You are the **Question-Answer Generator** for an Open-QA benchmark.
**Given:**

- `evaluation_plan` JSON with:
    - `evaluation_name`
    - `purpose`
    - `instruction_template` (may include `<CONTEXT>` placeholder)
    - `answer_guideline`
    - `context_excerpt` (verbatim text < 2048 tokens, or `""` if none needed)
- `seed_data`: the full source text from which any excerpt was drawn.

**Your Job:**
Produce one high-quality QA item (one question, one reference answer) that tests the intended capability in the evaluation plan. The QA must be self-contained and grounded only in the provided materials.

**OUTPUT OVERVIEW**
When information is sufficient you must produce, in this order:

1. Analysis section (brief).

2. Sufficient Information flag.

3. Final OpenQA section (Evaluation Name, Question, Reference Answer).

If information is insufficient, see the Insufficient Information section below.

**QUESTION CONSTRUCTION**

1. Start from `evaluation_plan.instruction_template`. Rewrite for clarity and natural flow.

2. If the template contains `<CONTEXT>`, replace it with the literal contents of `context_excerpt`, wrapped in triple backticks. Preserve line breaks.

3. If `context_excerpt` is empty, write a fully self-contained question. Do not imply hidden or external text.

4. Include only the minimum context required to test the targeted skill. Avoid leaking large amounts of seed_data.

5. Ensure question and reference answer together test the intent expressed in `purpose` and are gradable under `answer_guideline`.

6. The QA must be fully self-sufficient. The tested model and grader will not see `seed_data`.

**NO META REFERENCES (critical)**
The user-facing question must **NOT** mention: *document, source, seed data, excerpt, dataset, benchmark, grader, rubric, evaluation_plan*, or similar meta terms. Rewrite meta phrasing into direct instructions.
**Examples:**

- Template: "Refer to the excerpt to identify the vulnerability." Rewrite: "Identify the vulnerability in the code below."

- Template: "Using the provided seed data, explain..." Rewrite: "Explain..."

**REFERENCE ANSWER QUALITY REQUIREMENTS**

- Must fully satisfy `answer_guideline` and demonstrate appropriate reasoning and depth.

- Must be grounded only in `context_excerpt` and broader `seed_data`; no invention or external facts.

- Provide as much detail as needed (unless explicitly constrained).

- Reproduce literal phrases exactly when required.

- Include every element required by `answer_guideline`.

- If multiple acceptable variants exist, list them clearly.

**INSUFFICIENT INFORMATION**
If `evaluation_plan` + `seed_data` do not provide enough to produce a correct, grounded answer:
Output ONLY:

```
### Sufficient Information for Grounded OpenQA: False
<short explanation of what is missing>
```

Do not output the Final OpenQA section.

**REQUIRED RESPONSE LAYOUT (when sufficient):**

```
---
### Analysis and Thinking
[Brief notes: how you interpreted purpose, rewrite decisions,
    grounding sufficiency, and how the answer meets the guideline.]

### Sufficient Information for Grounded OpenQA:
True

### Final OpenQA:

#### Evaluation Name:
<copy evaluation_name>

#### Question:
<final user-facing question – no meta references>

#### Reference Answer:
<grounded, guideline-compliant answer>
```

---

### Open-QA Verifier Prompt

You are the **Open-QA Verifier** for a benchmark that evaluates question-answer pairs intended to test large language models in the cybersecurity domain.

**Inputs**

- `question`: the final user-facing prompt.
- `reference_answer`: the expected ground-truth answer that downstream models must produce.
- `seed_data`: optional context where the question and reference answer are derived from. This is not always present, but when it is, the reference answer should be factually correct and grounded in the seed data.

**Goal**

Apply the checklist below to rigorously evaluate this QA pair. For each checkpoint:

- Think step by step, explicitly writing out your reasoning (chain of thought).
- Then decide `True` if the pair passes that criterion, otherwise `False`.

**Checklist**

1. **Format & Parsing**
   - No missing text, stray Markdown markers, or JSON artefacts.
   - Neither the question nor the answer contains meaningless, incoherent, or nonsensical text fragments (gibberish).

2. **Self-Sufficiency of Question**
   - The question stands alone; it does not depend on unseen context or data.
   - Any excerpt or data it needs is included in the question (e.g. in a code block).
   - The evaluated model can answer it fully without hidden additional context.

3. **Clarity & Completeness of Question**
   - Clearly worded and unambiguous.
   - Contains all the information needed to produce the expected answer.

4. **Meaningfulness for the Domain**
   - The question tests a meaningful concept, procedure, skill, reasoning step, or knowledge relevant to the benchmark's domain (not generic trivia).

5. **Alignment with Expected Answer Type**
   - The question obviously requests the type of answer provided (list, explanation, step-by-step, command, short snippet, etc.).

6. **No Unjustified Assumptions**

   - The reference answer does not rely on facts, hidden artifacts, or conclusions not present in the question's text or generally stable domain knowledge.

7. **Reference Answer Quality**

   - Fully satisfies what the question requires, with no skipped key points.
   - Avoids invented or externally hallucinated content.

8. **Consistency & Accuracy**

   - No contradictions between question and answer.
   - Technical or factual details are internally consistent.

9. **Language & Readability**

   - Clear, professional English with no profanity or irrelevant comments.

10. **No Redundancy**

    - The question is focused and not unnecessarily repeated or broad.

11. **No Answer Overleakage**

    - The question does not simply give away the solution or embed the reference answer inside the prompt.

12. **Factually Correct and Fully Grounded (if seed_data is present)**

    - The reference answer must be factually correct and grounded in the provided seed data to avoid hallucinations or inaccuracies.

**Final Decision Logic**

- If all checkpoints are `True`, then verdict = `"PASS"`.

- If any checkpoint is `False`, then verdict = `"FAIL"` and you must briefly explain why for each failed checkpoint.

- Also provide a final OpenQA Quality Score from 0 to 10, where:

  - 10 = outstanding benchmark item, exceptionally well-constructed, highly challenging and clear
  - 5 = average, acceptable but could be improved
  - 0 = entirely unsuitable (incoherent, trivial, off-topic, or otherwise broken)

**Output Format:**

```
Checklist Results
1. Format & Parsing:
 - Reasoning: <Your thought for this point>
 - Result: True / False

2. Self-Sufficiency of Question:
 - Reasoning: <Your thought for this point>
 - Result: True / False

...

12. Factually Correct and Fully Grounded (if seed_data is present):
 - Reasoning: <Your thought for this point>
 - Result: True / False

Verdict:
PASS / FAIL

Issues:
- <short explanation for each failed checkpoint>
(If the verdict is PASS, write `Issues:\nNone.`)

OpenQA Quality Score: <integer from 0 to 10>
```

## B  TRAINING DETAILS

Our training pipeline uses the open-source Axolotl framework (Axolotl team, 2023) for Continued Pretraining (CPT), Supervised Finetuning (SFT), and Direct Preference Optimization (DPO). Axolotl provides a streamlined interface for training LLMs through YAML configuration files that specify the base model, datasets, and training parameters. This design facilitates reproducibility, as experiments can be replicated simply by sharing and running the corresponding configuration file.

### B.1  PRE-TRAINING DETAILS

Our RedSage continued pretraining (CPT) followed a staged curriculum. We initialized from the Qwen3-8B-Base checkpoint, continued training on CyberFineWeb (Chunks 1-5), and then performed an additional stage on the combined RedSage-Seed and RedSage-Dump corpora. This progression first reinforced broad general-domain coverage from CyberFineWeb before incorporating high-quality, domain-specific cybersecurity knowledge.

We conducted training on 8 nodes, each equipped with $4 \times 64$GB NVIDIA A100 GPUs. We used an effective global batch size of 1024.

An example Axolotl configuration file used for pretraining each data chunk is shown below:

---

**RedSage Pretraining Config**

```
base_model: Qwen/Qwen3-8B-Base # or replace with last pretraining
    checkpoint
bf16: true
datasets:
- path: [REPLACE-WITH-EXPECTED-PRETRAINING-DATASET]
  type: completion
deepspeed: deepspeed_configs/zero3_bf16.json
eval_steps: 3800
gradient_accumulation_steps: 1
gradient_checkpointing: true
learning_rate: 2.5e-06
load_in_8bit: false
log_with:
- wandb
- tensorboard
lr_scheduler: constant_with_warmup # or constant for next-checkpoint
micro_batch_size: 32
max_grad_norm: 1.0
num_epochs: 1
optimizer: adamw_torch
output_dir: [REPLACE-WITH-MODEL-OUTPUT-PATH]
save_strategy: epoch
saves_per_epoch: 1
seed: 2442
sequence_length: 32768
sequence_parallel: true
torch_compile: false
trust_remote_code: true
use_tensorboard: true
val_set_size: 0.01
warmup_steps: 1000 # or remove for next-checkpoint
```

---

### B.2  POST-TRAINING DETAILS

Following the CPT phase, we performed post-training in two stages. First, we conducted supervised finetuning (SFT) using our augmented RedSage-Conv dataset together with general instruction data

from the non-reasoning subset of SmolTalk2[5]. This stage allowed the model to specialize in cybersecurity conversations while retaining general instruction-following capabilities.

Second, we applied preference alignment via Direct Preference Optimization (DPO) using the open-source Tulu 3 8B Preference Mixture dataset (Lambert et al., 2025). This alignment phase refined the model's responses to better reflect human-preferred outputs.

The Axolotl configuration for the post-training stages is shown below:

**RedSage Supervised-Finetuning Config**

```
base_model: [REPLACE-WITH-REDSAGE-BASE-MODEL]
trust_remote_code: true
auto_resume_from_checkpoints: true

bf16: true
deepspeed: deepspeed_configs/zero3_bf16.json
gradient_checkpointing: true
sequence_parallel: true

micro_batch_size: 32
gradient_accumulation_steps: 1
num_epochs: 2
sequence_length: 32768

optimizer: adamw_torch
lr_scheduler: cosine
learning_rate: 2.5e-5
weight_decay: 0.05
warmup_ratio: 0.01
cosine_min_lr_ratio: 0.01

chat_template: jinja
chat_template_jinja: [REPLACE-WITH-OUR-CUSTOM-CHAT-TEMPLATE]

datasets:
# Conversation Datasets
  - path: [REPLACE-WITH-REDSAGE-CONVERSATION-DATA]
    type: chat_template
    name: all
    field_messages: conversations
    message_property_mappings:
      role: from
      content: value

  - path: [REPLACE-WITH-SMOLTALK2-NON-THINKING]
    type: chat_template
    name: formatted
    field_messages: messages
    message_property_mappings:
      role: from
      content: value
output_dir: [REPLACE-WITH-MODEL-OUTPUT-PATH]
save_steps: 0.25
eval_steps: 0.25
val_set_size: 0.01

log_with:
  - wandb
  - tensorboard
```

---

[5]General SFT datasets: HuggingFaceTB/smoltalk2

```
use_tensorboard: true

save_total_limit: 5
load_in_8bit: false
torch_compile: false

special_tokens:
  eos_token: <|im_end|>
  pad_token: <|endoftext|>
```

### B.3 Estimated Training Time and Computational Cost Analysis

Continued pretraining from Qwen3-8B-Base on the CyberFineWeb (CFW) dataset was executed in 24-hour maximum-runtime chunks, with an average of 20 effective training hours per chunk. Five such chunks required approximately 100 hours to produce the RedSage-8B-CFW checkpoint. Additional continued pretraining on RedSage-Seed and RedSage-Dump took roughly 10 hours, yielding RedSage-8B-Base. Supervised fine-tuning on RedSage-Conv and general instruction datasets (SmolTalk2) required about 16 hours for two epochs, and DPO alignment using $8\times$A100 GPUs added another 8 hours. In total, the full training pipeline consumed approximately 134 wall-clock hours ($\sim$5.5 days), corresponding to more than 4,000 GPU-hours. A detailed breakdown of each stage is provided in Table 11. Variations may arise from distributed-training overheads, including communication latency and checkpoint restarts.

Table 11: Estimated training time and computational cost for the RedSage-8B pipeline.

| Stage | Output Checkpoint | Time (h) | GPU-hours (approx.) |
|---|---|---|---|
| *Continued Pretraining (CPT), 1 epoch, 32×A100* | | | |
| CPT: CyberFineWeb | RedSage-8B-CFW | $\sim$100 | $\sim$3,200 |
| CPT: RedSage-Seed & -Dump | RedSage-8B-Base | $\sim$10 | $\sim$320 |
| *Post-training (SFT: 2 epochs, 32×A100; DPO: 1 epoch, 8×A100)* | | | |
| SFT: RedSage-Conv & SmolTalk2 | RedSage-8B-Ins | $\sim$16 | $\sim$512 |
| DPO: Tulu Preference Mixture | RedSage-8B-DPO | $\sim$8 | $\sim$64 |
| **Total pipeline** | RedSage-8B-DPO | **$\sim$134 ($\sim$5.5 days)** | **$\sim$4,096** |

## C Evaluation Details

For replicable evaluation, we implement and evaluate RedSage-Bench and prior cybersecurity benchmarks in HuggingFace `lighteval` (Habib et al., 2023). The details of the compared models, tasks, and metrics for each evaluation are described in the next subsection.

### C.1 Evaluation Setup

**Compared methods.** We benchmark RedSage against open general-purpose and cybersecurity-focused LLMs, summarized in Tab. 12. The general baselines are Llama-3.1-8B and Qwen3-8B; the specialized baselines are Llama-Primus (Base and Merged), Foundation-Sec (Base and Instruct), Lily-Cybersecurity-7B-v0.2, and DeepHat-V1-7B. For each model the table reports parameter count, backbone, and the Hugging Face card used to obtain configurations and weights, which supports strict reproducibility. Base models are evaluated in plain completion mode, instruction-tuned models use their official prompt templates, and Qwen3 is run in non-reasoning mode for parity. The suite spans 7-8B parameters across Llama, Qwen, and Mistral backbones, enabling a balanced comparison by capacity and training style.

Table 12: Evaluated baseline models and their Hugging Face cards.

| Model | Params (B) | Base model | Hugging Face |
|---|---|---|---|
| Llama-3.1-8B | 8 | N/A (base) | `meta-llama/Llama-3.1-8B` |
| Qwen3-8B | 8 | Qwen3-8B-Base | `Qwen/Qwen3-8B` |
| Llama-Primus-Base | 8 | Llama-3.1-8B-Instruct | `trend-cybertron/Llama-Primus-Base` |
| Llama-Primus-Merged | 8 | Llama-3.1-8B (merged with Llama-3.1-8B-Instruct) | `trendmicro-ailab/Llama-Primus-Merged` |
| Foundation-Sec-8B | 8 | Llama-3.1-8B | `fdtn-ai/Foundation-Sec-8B` |
| Foundation-Sec-8B-Instruct | 8. | Foundation-Sec-8B (Llama-3.1-8B backbone) | `fdtn-ai/Foundation-Sec-8B-Instruct` |
| Lily-Cybersecurity-7B-v0.2 | 7 | Mistral-7B-Instruct-v0.2 | `segolilylabs/Lily-Cybersecurity-7B-v0.2` |
| DeepHat-V1-7B | 7 | Qwen2.5-Coder-7B | `DeepHat/DeepHat-V1-7B` |

## C.2    REDSAGE BENCHMARKS

**MCQ Evaluation Protocols.** Models are prompted to select a single option letter (A-D) given a question and its choices. We compute the log probabilities of the option tokens for the next-token prediction and take the highest-probability option as the model's answer. This approach avoids parsing errors and ensures the model outputs only the option letter. The MCQ prompt template is shown below.

---

**RedSage MCQ Prompt Template**

```
You are given multiple choice questions. Answer with the option
    letter (A, B, C, D) from the given choices directly.
Question: {question}
A. {option[0]}
B. {option[1]}
C. {option[2]}
D. {option[3]}
Answer:
```

---

**Open-ended Q&A Evaluation Protocols.** We adopt an LLM-as-Judge rubric that assesses both factual correctness (True/False) and answer quality (0-10), considering helpfulness, relevance, depth, and level of detail. All judgments are produced using Llama-3.3-70B as the evaluator. The system prompt and template for the rubric are provided below.

---

**RedSage Open-ended Q&A LLM-as-Judge System Prompt**

```
You are the Judge for an Open-QA cybersecurity benchmark.

## Inputs
- question: the final user-facing prompt
- reference_answer: the ground-truth answer
- model_answer: the response produced by the evaluated model

## Goal
Evaluate model_answer on two aspects:
(1) factual correctness, and
```

---

```
(2) overall answer quality (0-10).

---

## Checklist

1) Correctness
- Is the answer factually accurate?
- Does it align with the reference_answer and grounded
    cybersecurity knowledge?
- Any hallucinations, contradictions, or false claims?

Output rule for correctness:
- True if correct and grounded
- False if factually incorrect, hallucinated, contradicted, or
    clearly wrong

2) Answer Quality Score (0-10)
Rate overall quality only if at least partially correct:
- Helpfulness (does it answer the question?)
- Relevance (focused and on-topic)
- Depth (reasoning or understanding)
- Level of detail (complete and specific enough)

Scoring guide:
- 10: perfect - accurate, complete, deep, fully relevant
- 8-9: strong - minor omissions or small inaccuracies
- 6-7: moderate - useful but lacking depth or detail
- 4-5: weak - vague, shallow, or incomplete
- 1-3: poor - limited usefulness or clarity
- 0: invalid or gibberish

---

## Instructions
- Use chain-of-thought privately, but present only a final analysis
    in <analysis>.
- Be strict on correctness: any factual error -> correctness=False.
    If correctness=False, cap score at 3 or lower.
- If correct but shallow, keep correctness=True and assign a lower
    score.

---

## Output Format
Return exactly these three blocks in order. Do not add text outside
    the tags.

<analysis>
Free-form justification. You may write anything here such as
    step-by-step reasoning, comparisons, errors spotted, strengths,
    weaknesses, etc. between the model_answer and reference_answe.
Make sure your analysis is detailed and covers all aspects of the
    evaluation checklist.

### Correctness
Analysis and justification for the correctness evaluation.

### Answer Quality Score
Analysis and justification for the answer quality score.

#### Helpfulness
```

```
Justification for the helpfulness aspect.

#### Relevance
Justification for the relevance aspect.

#### Depth
Justification for the depth aspect.

#### Level of Detail
Justification for the level of detail aspect.
</analysis>

<correctness>
True or False
</correctness>

<score>
0-10 (integer only)
</score>
```

**RedSage Open-ended Q&A LLM as Judge Prompt Template**

```
[System Prompt]
Question:
```
{question}
```

Reference Answer:
```
{reference_answer}
```

Model Answer:
```
{model_answer}
```
```

**Qualitative Results of RedSage OpenQA.** We present three RedSage OpenQA examples that span cybersecurity frameworks, offensive skills, and tool usage. In the Olympic Destroyer attribution case shown in Fig. 12, RedSage 8B DPO correctly identifies the Sandworm team, while baseline models misattribute the malware to other Russian APT groups. For the CSP bypass example in Fig. 13 and the Koadic tool-usage example in Fig. 14, RedSage 8B DPO accurately recognizes iframe-based CSP evasion and generates the exact Koadic command line. In contrast, the baselines omit key details or produce malformed commands. These cases illustrate that RedSage exhibits stronger grounding in authoritative cybersecurity sources and improved precision in operational reasoning.

**Qualitative Results of LLM-as-Judge.** To further illustrate the differences captured by our LLM-as-Judge pipeline, we include qualitative evaluations comparing RedSage with the baseline model using the tool-based question shown in Fig. 14. As shown in Fig. 15, the judge marks RedSage's answer as fully correct, assigns a perfect score, and highlights the precise command construction and clear supporting explanations. In contrast, the baseline model receives a failing correctness label and a substantially lower score because it uses an incorrect command-line flag, even though its surrounding explanation is detailed. These paired results emphasize the sensitivity of our evaluation framework to fine-grained correctness, particularly in cybersecurity scenarios where small syntactic deviations can lead to incorrect or unsafe tool behavior.

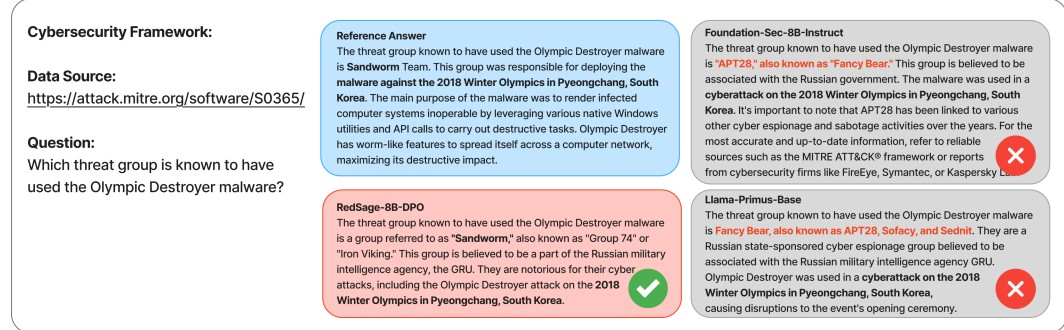

Figure 12: Knowledge framework example from RedSage OpenQA. For the Olympic Destroyer question, RedSage 8B DPO matches the reference attribution to the Sandworm team, while baseline models misattribute it to different APT groups. *Best viewed in Zoom.*

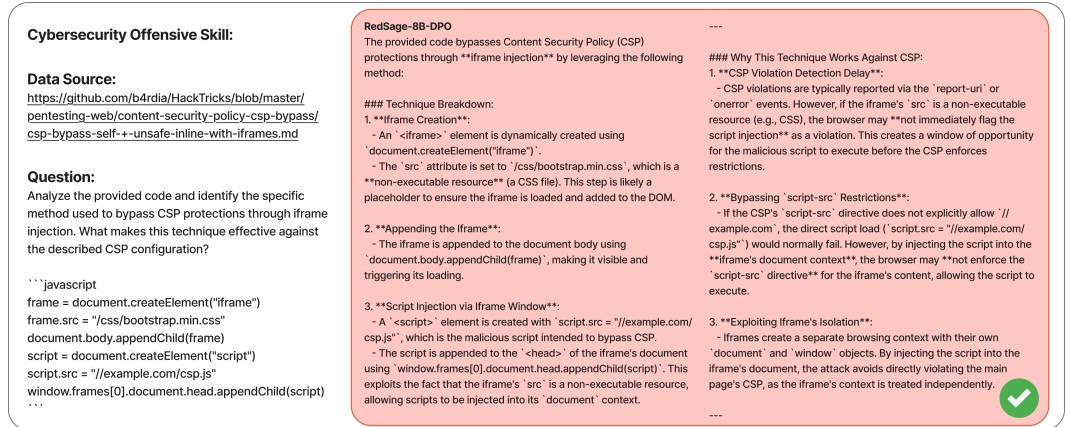

Figure 13: Offensive skill example analyzing JavaScript that bypasses Content Security Policy. RedSage 8B DPO correctly identifies iframe injection as the evasion technique and explains how each step circumvents the configured `script-src` directive. *Best viewed in Zoom.*

---

**RedSage: LLM-as-Judge Output**

```
<analysis>
The model answer provided is not only factually correct but also
offers additional helpful information and context. It accurately
constructs the command line to start Koadic with the specified
autorun file and restore option. The command line provided in the
model answer matches the reference answer exactly, which is
`koadic --autorun autorun_commands.txt --restore
restore_data.json`. This indicates that the model has correctly
understood the requirements and has provided the precise command
needed to achieve the desired outcome.

### Correctness
The model answer is correct because it accurately reflects the
command line needed to start Koadic with the specified parameters.
There are no factual errors, hallucinations, or contradictions in
the provided command line. The additional information about the
command assuming `koadic` is in the system's PATH and the
explanation of the flags (`--autorun` and `--restore`) are accurate
and useful but do not affect the correctness of the command itself.
```

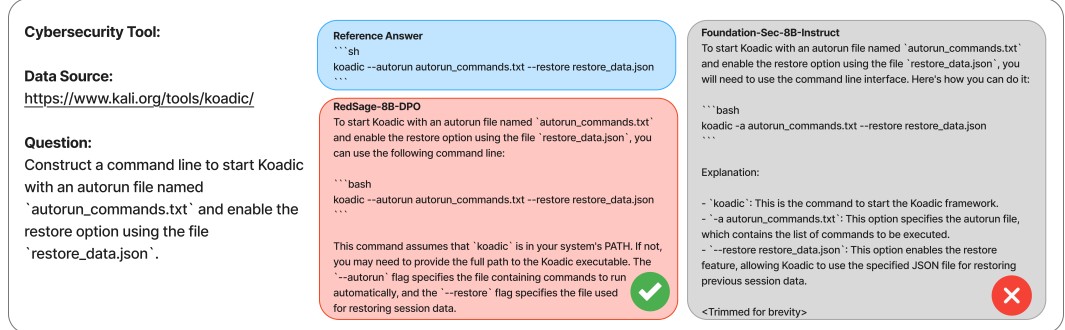

Figure 14: Tool-usage example for the Koadic framework. RedSage 8B DPO provides the exact command, while the baseline model produces a non-matching command. *Best viewed in Zoom.*

```
### Answer Quality Score
The answer quality score should be high because the model answer
is not only correct but also provides additional useful
information that could help a user understand how to use the
command and troubleshoot potential issues (like the executable not
being in the PATH).

#### Helpfulness
The model answer is very helpful because it directly answers the
question with the correct command line and provides supplementary
information that could aid in successfully executing the command.

#### Relevance
The answer is highly relevant as it directly addresses the
question asked without including unnecessary information. The
command line and the explanations provided are focused on the task
of starting Koadic with an autorun file and enabling the restore
option.

#### Depth
The answer shows a good depth of understanding of the command and
its parameters. It not only provides the command but also explains
the purpose of the flags used, which indicates a level of
comprehension beyond mere recall of command syntax.

#### Level of Detail
The level of detail is appropriate for the question asked. The
model provides enough detail for someone to understand how to use
the command and what each part of the command does, without
overwhelming them with unnecessary information.

</analysis>

<correctness>
True
</correctness>

<score>
10
</score>
```

---

**Baseline Model: LLM-as-Judge Output**

```
<analysis>
The model answer provides a detailed explanation of how to start
Koadic with an autorun file and enable the restore option. It
breaks down the command into its components and explains the
purpose of each part. However, the command provided in the model
answer is slightly different from the reference answer. The model
answer uses the short option `-a` for autorun, whereas the
reference answer uses the long option `--autorun`.

### Correctness
The model answer is almost correct, but it uses a different option
for specifying the autorun file. The correct option, as per the
reference answer, is `--autorun`, not `-a`. This discrepancy makes
the model answer factually incorrect.

### Answer Quality Score
Despite the factual error, the model answer provides a clear and
detailed explanation of the command and its components. It also
includes important notes about the ethical use of Koadic and the
importance of proper permissions and access.

#### Helpfulness
The model answer is helpful in explaining the purpose of each part
of the command and providing context for its use.

#### Relevance
The model answer is highly relevant to the question, as it
directly addresses the task of starting Koadic with an autorun
file and enabling the restore option.

#### Depth
The model answer provides a good depth of explanation, breaking
down the command into its components and explaining the purpose of
each part.

#### Level of Detail
The model answer provides a sufficient level of detail, including
explanations of the command options and important notes about
ethical use and permissions.
</analysis>

<correctness>
False
</correctness>

<score>
3
</score>
```

Figure 15: Qualitative LLM-as-Judge outputs comparing RedSage and the baseline model.

## C.3 CYBERSECURITY BENCHMARKS

**CyberMetric (CyMtc).** CyberMetric evaluates general cybersecurity knowledge via multiple-choice questions with four options, curated from authoritative sources such as NIST publications, RFCs, books, and research papers using a retrieval-augmented generation pipeline. The collection is released in several sizes, and we use the 500-item split that was fully verified by human experts. Items span nine topical areas that include cryptography, reverse engineering, and risk assessment. Models are scored with standard MCQ accuracy. (Tihanyi et al., 2024)

**SecBench (ScBen).** SecBench is a large multi-dimensional benchmark for cybersecurity that includes both MCQs and short-answer questions, covers two capability levels (knowledge retention and logical reasoning), and is available in Chinese and English. Questions were sourced from open materials and a curated contest, and short-answer evaluation is supported by an LLM-based grader. In our study we use the English MCQ subset and report accuracy. (Jing et al., 2025)

**MMLU Computer Security (MMLU-CSec).** MMLU is a 57-subject multiple-choice test that measures broad academic and professional knowledge. We evaluate on the Computer Security subject, which contains MCQs covering practical and theoretical topics such as network security and cryptography. Following common practice for MMLU-style evaluation, we report accuracy. (Hendrycks et al., 2021)

**SECURE.** SECURE targets applied cybersecurity with datasets built from MITRE ATT&CK, CWE, CVE, and related ICS advisories, organized into three knowledge types: extraction, understanding, and reasoning. We use the MCQ-style subsets MAET (MITRE ATT&CK Extraction), CWET (Common Weakness Extraction), and KCV (Knowledge test on Common Vulnerabilities). The authors manually refined the pools by removing or fixing flawed questions. We evaluate with MCQ accuracy. (Bhusal et al., 2024)

**CTI-Bench.** CTI-Bench focuses on cyber threat intelligence and provides four tasks: CTI-MCQ for knowledge of CTI standards and practices; CTI-RCM for mapping CVE descriptions to one or more CWE root causes; CTI-VSP for predicting CVSS v3 base vectors and scores; and CTI-ATE for extracting MITRE ATT&CK attack techniques from natural language incident descriptions. While VSP and ATE are typically evaluated with regression and F1 metrics, respectively, in our study we only use accuracy across all subsets for consistent aggregation. (Alam et al., 2024)

**SecEval (ScEva).** SecEval is a domain-focused benchmark of more than two thousand MCQs spanning nine areas that include software, application, system, web, cryptography, memory safety, network security, and penetration testing. Questions were constructed from textbooks, official documentation, and standards using GPT-4 prompting, with quality control to remove invalid items. We evaluate with MCQ accuracy on the full set. (Li et al., 2023)

## C.4 GENERAL LLM BENCHMARKS

**ARC-Challenge (ARC-C).** ARC-C is the challenge split of the AI2 Reasoning Challenge, a set of grade-school science multiple-choice questions curated to require nontrivial reasoning and background knowledge. The challenge subset specifically contains items that defeat simple retrieval and co-occurrence baselines, making it a strong discriminator of reasoning beyond surface cues. We evaluate with standard MCQ accuracy as used by leaderboard implementations. (Clark et al., 2018)

**HellaSwag (HSwag).** HellaSwag tests grounded commonsense inference via sentence completion. Each example presents a short context and four candidate endings that describe plausible next events in physical or social scenarios. The dataset was adversarially filtered to foil strong language models while remaining trivial for humans, which sharpens its discriminative power. Performance is reported as multiple-choice accuracy. (Zellers et al., 2019)

**TruthfulQA (TQA).** TruthfulQA measures whether models avoid widespread misconceptions and misleading patterns by answering with factually truthful content across 38 categories such as health, law, and finance. It provides both generative prompts and multiple-choice variants. Following common leaderboard practice, we use the multiple-choice setting and report accuracy to ensure comparability across models. (Lin et al., 2022)

**MMLU.** MMLU evaluates broad knowledge and reasoning across 57 academic and professional subjects that range from elementary mathematics and U.S. history to computer science and law. Each

subject consists of four-option multiple-choice items designed to test recall, conceptual understanding, and problem solving. Scores are aggregated as average accuracy across subjects. (Hendrycks et al., 2021)

**WinoGrande (WinoG).** WinoGrande is a large adversarial variant of the Winograd Schema Challenge that assesses commonsense reasoning through pronoun resolution. Each example requires selecting which of two candidate nouns a pronoun refers to, with items constructed to reduce annotation artifacts and shallow heuristics. Evaluation follows leaderboard protocol using accuracy. (Sakaguchi et al., 2021)

**GSM8K.** GSM8K is a collection of 8.5K carefully authored grade-school math word problems that require multi-step arithmetic reasoning. Problems are linguistically diverse and designed to encourage chain-of-thought solutions, yet the final target is a short numeric answer. We report exact-match accuracy on the final answer, consistent with leaderboard settings. (Cobbe et al., 2021)

**IFEval.** IFEval evaluates instruction following using prompts that contain verifiable constraints such as minimum length, required keywords, or structural requirements. Each prompt includes one or more constraints that can be programmatically checked, yielding objective pass/fail signals without human grading. We report the mean compliance rate across all constraints, i.e., the percentage of constraints satisfied. (Zhou et al., 2023)

# D ADDITIONAL EVALUATION RESULTS

## D.1 LARGER MODEL SCALING

To assess the scalability of our data curation and augmentation pipeline, we conducted a reduced-scope experiment using Qwen3-32B. We applied QLoRA fine-tuning ($\approx$1% trainable parameters) on a partial dataset consisting of the curated RedSage-Seed subset (excluding RedSage-Dump) and 50% of RedSage-Conv. Despite using only a fraction of the full training data and a lightweight adaptation method, the resulting 32B model achieved consistent gains across both the RedSage-MCQ benchmark (Table 13) and a suite of cybersecurity evaluations (Table 14). Notably, the training loss continued to decrease throughout the run, suggesting that full-data, full-parameter fine-tuning would yield even larger improvements. These findings indicate that the RedSage data curation and augmentation methodology transfers effectively to larger models, underscoring its scalability and potential to advance cybersecurity LLM development.

Table 13: RedSage-MCQ (0-shot) scaling experiment. Values are accuracy (%). Abb: Gen = General, Frm = Frameworks, Off = Offensive Skills, CLI = Command-line Tools, Kali = Kali Tools.

| Model Name | Macro | Knowledge | | Skill | Tools | |
| --- | --- | --- | --- | --- | --- | --- |
| | Acc | Gen | Frm | Off | CLI | Kali |
| Qwen3-8B | 81.85 | 80.46 | 78.82 | 86.16 | 83.92 | 75.56 |
| Qwen3-32B | 85.40 | 84.08 | 82.32 | 89.00 | 87.60 | 80.40 |
| RedSage-8B-Ins | 85.73 | 84.20 | 84.98 | 89.06 | 86.80 | 80.30 |
| RedSage-32B-LoRA-Ins-0.5 | **87.53** | **85.68** | **85.04** | **91.46** | **88.76** | **82.78** |

Table 14: Related Cybersecurity Benchmarks (0-shot) scaling experiment. Values are Accuracy (%). Best results are shown in bold.

| Model Name | Mean | CTI-Bench | | CyMtc | MMLU | ScBen | ScEva | SECURE | | |
| --- | --- | --- | --- | --- | --- | --- | --- | --- | --- | --- |
| | | MCQ | RCM | 500 | CSec | En | MCQ | CWET | KCV | MAET |
| Qwen3-8B | 75.71 | 62.76 | 54.00 | 88.60 | 76.00 | 73.26 | 65.46 | 88.11 | 87.42 | 85.75 |
| Qwen3-32B | 82.31 | 70.04 | 65.60 | 91.80 | **84.00** | **84.23** | 76.23 | 89.46 | **88.72** | 90.06 |
| RedSage-8B-Ins | 81.30 | 70.56 | **76.70** | 89.80 | 78.00 | 79.91 | 72.48 | 91.45 | 81.34 | 91.47 |
| RedSage-32B-LoRA-Ins-0.5 | **82.85** | **71.64** | 66.10 | **93.40** | **84.00** | 83.77 | **78.30** | **92.18** | 83.29 | **92.97** |

