# OpenReview forum: "RedSage: A Cybersecurity Generalist LLM"
_ICLR.cc/2026/Conference — ICLR 2026 Poster_

### Official Review · Reviewer_QvXR · 2025-10-23

**Soundness:** 3
**Presentation:** 3
**Contribution:** 4
**Rating:** 6
**Confidence:** 4

**Summary:**

This paper introduces RedSage, an open-source cybersecurity-oriented LLM trained via domain-aware continual pretraining and agentic post-training augmentation. This work also proposes RedSage-Bench, a 30K-question benchmark spanning a diverse range of tasks. Authors of this paper claim the model outperforms state-of-the-art specialized baselines on both cybersecurity benchmarks and general LLM tasks, demonstrating that cybersecurity specialization can enhance both domain-specific and general reasoning.

**Strengths:**

Clear presentation: This work integrates all stages (CPT, SFT, DPO) with substantial data at each phase, which is clearly shown in Fig. 1. Readers can have a very straightforward view on how RedSage was trained on the dataset selected.

Comprehensive Training data coverage: The integration of CyberFineWeb, curated RedSage-Seed, and agentic augmentation provides strong coverage across cybersecurity subfields.

Proposed New Benchmark: What I am interested in is that this work expands prior benchmarks by incorporating tool proficiency and qualitative evaluation, from reviewer’s point of view, this innovation can close a key gap in cybersecurity LLM assessment.

**Weaknesses:**

May need computational cost analysis: While "compute constraints" are briefly mentioned in page 4 (CyberFineWeb section), there's no breakdown of training time, GPU-hours, or carbon footprint across stages. Adding some analysis on computational cost would help readers to form a general impression on the scale of RedSage training process.

Limited human validation: One of my concern on this work is the data part is heavily reliance on LLM-based verification, which could introduce subtle self-reinforcing biases.

Teacher model (verifier) analysis: Augmentation uses only Llama-3.3-70B and Qwen2.5-72B. Analysis on teacher model can make this paper more solid, such as examining sensitivity to teacher model choice, temperature settings, or comparison with smaller/different teacher models.

Minor issue: Font size on Fig. 3 and Fig. 5 can be larger which I felt difficulty to read

**Questions:**

As mentioned in the weakness, how do you ensure LLM-generated augmentation does not reinforce factual errors or tool misuse patterns?

How would RedSage perform in interactive cybersecurity reasoning task such as CTF agentic settings on benchmarks like CyBench?

Does the author consider data contamination for the proposed datasets and benchmark?

**Details Of Ethics Concerns:**

I think it would be a minor issue as authors of this paper already stated in the ethics statement section.

---

> ### Author Response · Authors · 2025-11-21
> **Reply to Reviewer QvXR: Part1**
>
> We thank the reviewer for the encouraging and insightful comments. Please find our responses to specific queries below.
>
> ### 1: Estimated training time and computational cost analysis.
> Thank you for the feedback. Our model was trained using **32×A100 GPUs** distributed across **8 compute nodes** (4 GPUs per node) with a **global batch size of 1024**. The **CyberFineWeb (CFW)** dataset was divided into chunks with a maximum runtime of 24 hours each, resulting in approximately **20 hours per chunk**. Training for five chunks required about **100 hours (~4 days)** to complete the **RedSage-8B-CFW** model.
>
> Continuing pretraining on the **RedSage-Seed** and **CyberSage-Dump** data required an additional **10 hours**, producing **RedSage-8B-Base**. The **supervised fine-tuning (SFT)** stage on **RedSage-Conv** and general instruction data took roughly **16 hours** for two epochs, and the **DPO alignment** steps with **8×A100 GPUs** required about **8 hours**.
>
> Overall, the full training pipeline required approximately **134 wall-clock hours (~5.5 days)**, corresponding to **over 4,000 GPU-hours** in total compute. Variations may occur due to distributed training overheads such as communication delays or checkpoint restarts.
>
> We will include these training time and cost analysis details in the **Appendix** of the revised version. Given the substantial computational cost of this research, we believe that releasing all verified model checkpoints will greatly benefit the community and cybersecurity researchers, enabling access to high-quality domain-specialized LLMs that would otherwise be prohibitively expensive to reproduce.
>
> ---
>
> ### 2: Clarification on limited human validation and heavy reliance on LLM-based verification.
> **Q1: How to ensure LLM-generated augmentation does not reinforce factual errors or tool misuse patterns.**
>
> Thank you for the thoughtful question. We acknowledge in the **Limitation section** that while an LLM-based approach scales efficiently, it may still propagate biases or inaccuracies inherent to LLM generation. To minimize potential factual errors and tool misuse patterns while maintaining scalability, we designed a **multi-stage verification pipeline** as described in **L284–L290**.
>
> As detailed in **Section 3.3** and **Appendix A.4**, the first filtering stage focuses on **structural validity**, where a strong verifier LLM with chain-of-thought prompting evaluates multiple checklist criteria such as *format, correctness, distractors, topical relevance,* and *consistency* (see pp. 31–32 for the prompt template). Only data that pass all checklist items from **two independent verifier LLMs** proceed to the next phase.
>
> In the second stage, the verifier LLM performs **quality scoring** (see pp. 36–37 for the prompt template), assigning a score from 0 to 10. The data are then sorted by quality, and only the top-scoring samples are retained for random sampling. Importantly, the **open-ended QA subset** is human verified, and to further validate our **MCQ benchmark**, we randomly sample a subset of items for **manual review** to ensure that all filtered examples are factually correct and well-structured.
>
> This **hybrid verification approach**, combining LLM-based filtering with targeted human validation, is also used in related works such as **CyberMetric (Tihanyi et al., 2024)**, where only a partial subset of the LLM-generated benchmark is human verified. Despite the potential limitations of LLM-generated data, our post-trained **RedSage** models demonstrate **consistent improvements** across all benchmarks, including those with human-verified evaluations. This suggests that any residual bias or potential inaccuracy has minimal impact on performance and that our data generation process meaningfully enhances downstream results.
>
> Our findings align with prior studies such as **AgentInstruct (Mitra et al., 2024)**, which show that high-quality synthetic data can consistently improve benchmark performance. Moreover, LLM-generated synthetic data are widely adopted in modern training pipelines, including the **Qwen3** model itself (Yang et al., 2025).
>
> **References:**
> - Norbert Tihanyi et al. *CyberMetric: A Benchmark Dataset Based on Retrieval-Augmented Generation for Evaluating LLMs in Cybersecurity Knowledge.* IEEE CSR 2024, pp. 296–302. doi: 10.1109/CSR61664.2024.10679494.
> - Arindam Mitra et al. *AgentInstruct: Toward Generative Teaching with Agentic Flows.* arXiv:2407.03502, 2024.
> - An Yang et al. *Qwen3 Technical Report.* arXiv:2505.09388, 2025.
>
> ---

---

> ### Author Response · Authors · 2025-11-21
> **Reply to Reviewer QvXR: Part2**
>
> ### 3: Clarification on teacher model (verifier) selection and analysis of sensitivity to model choice and generation settings.
>
> We used **Llama-3.3-70B** and **Qwen2.5-72B**, the strongest open-source instruction-tuned models that could run locally with our compute resources during the data creation phase. In the **agentic data augmentation** stage, we iteratively refined prompts and verified outputs until they consistently produced high-quality **supervised fine-tuning**, **benchmark**, and **verification** data.
>
> Temperature varied by task: the default (0.7) encouraged diversity while maintaining grounded results for **post-training augmentation**, while a lower setting (0.3) improved accuracy and consistency during **data filtering and verification**.
>
> For validation, both **Llama-3.3-70B** and **Qwen2.5-72B** served as verifiers. An item was accepted only if **both models independently voted “pass,”** reducing false positives and improving reliability.
>
> To assess sensitivity to model choice and temperature, we re-verified 100 random samples. **Qwen2.5-72B** at 0.3 temperature matched our reference labels most closely. Raising the temperature from **0.3 to 1.5** caused up to a **4%** drop in F1 score, a modest decline likely mitigated by **chain-of-thought prompting**, which guided structured reasoning before final judgments.
>
> Using a smaller model (**Qwen3-8B**) lowered **precision** and **recall** to **86%** and **96%**, respectively. We will include these quantitative results and additional qualitative examples, such as **verification outputs**, **reasoning traces**, and **score distributions**, in the revised **Appendix** to illustrate verification behavior in detail.
>
>
> ---
>
> ### 4: Minor issue regarding figure font size.
> Thank you for the feedback. We will revise **Figures 3 and 5** to increase the font size and improve overall readability in the revised version.
>
> ---
>
> ### Q2. Clarification on RedSage performance in agentic cybersecurity tasks such as CyBench.
> Thank you for the question. Our current goal is to develop a **strong, locally deployable cybersecurity foundation model** that maintains **general reasoning capabilities** while supporting assistant-style use cases. This direction is motivated by the impracticality of relying on proprietary models for cybersecurity operations or assistance, since they may expose sensitive information to third parties.
>
> Our current evaluation focuses on assessing the model’s **internal reasoning and knowledge capabilities** rather than agentic behaviors. As described in the **Related Works** section, we exclude **CyBench** and **NYU-CTF** from evaluation because these benchmarks primarily assess **autonomous agent performance** rather than the LLM itself. The **RedSage data augmentation pipeline** is designed to simulate **realistic cybersecurity assistant conversations** and is not yet optimized for **autonomous, tool-interacting agentic workflows**.
>
> It is also worth noting that existing evaluations on **agentic cybersecurity benchmarks** such as **CyBench** have only been conducted with **large-scale models** deployed on AI servers, including **GPT-4**, **Claude**, and **Gemini**. The smallest open-source models tested, such as **LLaMA-70B** and **Mixtral-8×22B**, achieved only **5–7.5% accuracy** on unguided CyBench tasks and **0%** on NYU-CTF.
>
> We consider these **agentic reasoning tasks** an important direction for future research. The current **RedSage models** serve as a **practical and open foundation** toward that goal and represent a meaningful milestone for the future development of **autonomous cybersecurity agents**.
>
> ---
>
> ### Q3. Consideration on data contamination in the proposed datasets and benchmark.
>
> Thank you for raising this important question. We have carefully considered the possibility of **data contamination**. Our proposed benchmarks consist of **MCQ** and **open-ended question** formats, while our augmented post-training data are structured as **multi-turn conversations**. These are produced through **separate generation pipelines**, ensuring that no benchmark data are used during model training.
>
> To further prevent overlap, we apply an additional **data filtering and deduplication process**. Specifically, any synthetic post-training data whose query has a **semantic similarity greater than 0.9** to any benchmark question is removed. This procedure eliminates approximately **2.96% of training data relative to the benchmark size** (or **0.31% relative to the total training data size**).
>
> Furthermore, the **consistent improvements observed** across related cybersecurity benchmarks demonstrate that our trained model **generalizes well** to different cybersecurity data distributions and tasks.

---

### Official Review · Reviewer_f3PV · 2025-11-01

**Soundness:** 3
**Presentation:** 4
**Contribution:** 4
**Rating:** 8
**Confidence:** 3

**Summary:**

This paper presents RedSage, a cybersecurity-focused LLM that is trained with curated pre-training and post-training for cybersecurity knowledge. It presents an elaborate cybersecurity data curation pipeline that uses LLMs for isolating cybersecurity relevant data and augmenting multi-turn conversational data for post-training. The results demonstrate how both pre-training and post-training can help improve performance on cybersecurity tasks.

**Strengths:**

- The paper clearly demonstrates that cybersecurity curated pre- and post-training improves performance significantly without catastrophic forgetting of general knowledge
- The cybersecurity-specific curated dataset is incredibly useful for future training tasks

**Weaknesses:**

- The method for open-ended Q&A evaluation is not adequately described. Line 317 mentions "prefix exact match or regex matching" and points to Appendix C.1 however neither the text nor the appendix provide sufficient details or references to clearly understand this evaluation.
- The method for generating the instruction-tuned variant is not explained or referenced. A diagramatic view of how each of the variants was derived would be helpful.

**Questions:**

- Why is there a large imbalance between MCQ and open-ended Q&A (30K vs 240)? Is it because of human verification of open-ended benchmark? Please provide details of the human verification.
- Some estimates of training time would help future research aiming to replicate this work
- Were other LLMs tried for the agentic augmentation and why were Llama-3.3-70B and Qwen2.5-72B chosen?
- It would be interesting to see how RedSage fairs against models of higher size and commercial models atleast on the general benchmarks. For instance, RedSage-8B-DPO catches up with GPT4 performance on CTIBench-MCQ.

---

> ### Author Response · Authors · 2025-11-21
> **Reply to Reviewer f3PV: Part1**
>
> We thank the reviewer for the encouraging and insightful comments. Please find our responses to specific queries below.
>
>
> ### 1: Clarification on open-ended Q&A evaluation and “prefix exact match or regex matching” in L317.
>
> We thank the reviewer for pointing this out. The open-ended Q&A is evaluated using an **LLM-as-Judge rubric** that measures both **factual correctness (True/False)** and **answer quality (0–10)** across *helpfulness, relevance, depth,* and *level of detail* between the model’s response and the reference answer grounded in the seed data (L280–L282). We will explicitly describe this evaluation procedure, including the judge prompt template, and clarify the differences between **data generation** and **evaluation** in the revised **Appendix A.4**, after “Open-ended QA Benchmarks.”
>
> In Line 317, the phrase **“prefix exact match or regex matching”** refers to the evaluation setup for instruction-tuned models and structured output tasks. For example, in MCQs, the instruction-tuned model outputs only the option letter (A, B, C, or D). Using prefix exact match, outputs like “B. Option Answer” are still considered correct when the ground truth is “B.” For structured outputs such as **CTI-Bench RCM (Root Cause Mapping)**, where the model maps CVEs to CWE root causes (e.g., “CWE-XXX”), we use **regex matching** to robustly extract valid identifiers.
>
> We will update the appendix to specify the extraction method used for each benchmark and release our **evaluation code** for full transparency and reproducibility.
>
> ---
>
> ### 2: Clarification on the generation process of instruction-tuned variants and a suggestion to include a model derivation diagram.
>
> Thank you for the valuable feedback. The instruction-tuned variants are concisely described in the paper (L327–L330):
>
> > “We further derive instruction-tuned variants: RedSage-8B-Ins (instruction-tuned from Base) and the final RedSage-8B-DPO, which combines all data and applies DPO alignment.”
>
> Their data composition and training procedure are detailed in Section 3.4 (L304–L307):
>
> > “After pre-training, we fine-tune the base model on RedSage-Conv and general SFT data (Sec. 3.2) using a cosine learning rate schedule and apply direct preference optimization (DPO).”
>
> We agree that adding a clear diagram illustrating the training stages of each model variant would improve clarity. In the revised version, we will include a schematic showing the full derivation process:
>
> **[CFW] → (RedSage-8B-CFW) → [RedSage-Seed + Dump] → (RedSage-8B-Base) → [Conv + General SFT] → (RedSage-8B-Ins) → [Open Alignment Data] → (RedSage-8B-DPO)**
>
> where **[]** indicate data sources and **()** denote model variants. This figure will be added in the revision to clearly illustrate model lineage.
>
> ---
>
> ### Q1: Clarification of large imbalance between MCQ and open-ended Q&A and the human verification process.
>
> That is correct. The open-ended QA subset involves **human verification (L290)**, which does not scale as efficiently as **LLM-based verification** used for MCQs. The goal of open-ended QA is to evaluate **answer quality** in addition to correctness, across *helpfulness, relevance, depth,* and *level of detail*.
>
> Unlike MCQs, where only one correct answer exists, open-ended questions are inherently more complex to verify. For example, in *“List three types of …”*, the reference answer may be accurate, but other valid answers could exist. To handle such ambiguity, we manually reviewed and filtered these items to ensure **completeness and fairness**. Hence, the open-ended benchmark prioritizes **quality over quantity**, while MCQs already ensure large-scale quantitative coverage.
>
> The **240 open-ended QA items** provide sufficient evaluation coverage. For context, **MT-Bench** (Zheng et al., 2023), an early LLM-as-Judge benchmark, includes only **80 open-ended QA pairs**. Expanding our set further would be costly, since open-ended QA requires longer generations and **LLM-based judging**, unlike MCQs that only require selecting one option.
>
> Our human verification follows the same checklist as the verifier LLM used in Stage 2 scoring (L285–L290), with the full prompt in **Appendix A.4 (L1917–L1971)**. The checklist covers *format and parsing, clarity and completeness, domain relevance, expected answer alignment, factual correctness,* and *grounding.* Only items with verifier scores above 8 proceed to manual review, where annotators check edge cases such as alternative valid answers or borderline factual consistency.
>
> We will expand the description of the **human verification protocol** and the **dataset scale justification** in the revision.
>
> **Reference:**
> Zheng, L., Chiang, W. L., Sheng, Y., Zhuang, S., Wu, Z., Zhuang, Y., ... & Stoica, I. (2023). *Judging LLM-as-a-Judge with MT-Bench and Chatbot Arena.* *Advances in Neural Information Processing Systems*, 36, 46595–46623.

---

> > ### Author Response · Authors · 2025-11-21
> > **Reply to Reviewer f3PV: Part2**
> >
> > ### Q2: Estimated training time and computational cost analysis.
> >
> > Thank you for the feedback. Our model was trained using **32×A100 GPUs** distributed across **8 compute nodes** (4 GPUs per node) with a **global batch size of 1024**. The **CyberFineWeb (CFW)** dataset was divided into chunks with a maximum runtime of 24 hours each, resulting in approximately **20 hours per chunk**. Training for five chunks required about **100 hours (~4 days)** to complete the **RedSage-8B-CFW** model.
> >
> > Continuing pretraining on the **RedSage-Seed** and **CyberSage-Dump** data required an additional **10 hours**, producing **RedSage-8B-Base**. The **supervised fine-tuning (SFT)** stage on **RedSage-Conv** and general instruction data took roughly **16 hours** for two epochs, and the **DPO alignment** steps with **8×A100 GPUs** required about **8 hours**.
> >
> > Overall, the full training pipeline required approximately **134 wall-clock hours (~5.5 days)**, corresponding to **over 4,000 GPU-hours** in total compute. Variations may occur due to distributed training overheads such as communication delays or checkpoint restarts.
> >
> > We will include these **training time and cost analysis details** in the **Appendix** of the revised version. Given the substantial computational cost of this research, we believe that releasing all verified model checkpoints will **greatly benefit the community and cybersecurity researchers**, enabling access to **high-quality domain-specialized LLMs** that would otherwise be prohibitively expensive to reproduce.
> >
> > ---
> >
> > ### Q3: Clarification on the choice of LLMs for agentic augmentation.
> >
> > We used **Llama-3.3-70B** and **Qwen-2.5-72B** because they were the strongest open-source instruction-tuned models available that we could run locally with our compute resources during the data creation phase. We prioritized open-source models to enable **large-scale, iterative experimentation** and **continuous prompt refinement** without API cost constraints.
> >
> > We also conducted exploratory trials with **GPT-4o**, which can produce slightly higher-quality augmentations. However, since we focused on fully local, reproducible experimentation using our available compute infrastructure, we chose to proceed with **Llama-3.3-70B** and **Qwen-2.5-72B**. The consistent performance improvements achieved with these models demonstrate that they are sufficient to generate **high-quality conversational and evaluation data efficiently**.
> >
> > It would be valuable to extend our **agentic augmentation pipeline** with newer or larger models in future work. We are committed to **releasing our code**, allowing the community to further explore and enhance this pipeline with more recent LLMs.

---

> > > ### Author Response · Authors · 2025-11-21
> > > **Reply to Reviewer f3PV: Part3**
> > >
> > > ---
> > >
> > > ### Q4. RedSage performance comparison with higher-capacity and commercial models.
> > >
> > > Thank you for the positive comment and suggestion. We have added **Qwen3-32B** and **GPT-5** as higher-capacity and commercial baselines (evaluated in non-reasoning mode) in the updated benchmark tables below.
> > >
> > >
> > >
> > > #### **Our RedSage Benchmark**
> > >
> > > | Model | Macro Acc | Knowledge Gen | Knowledge Frm | Skill Off | Tools CLI | Tools Kali |
> > > |--------|-----------:|---------------:|---------------:|-----------:|-----------:|------------:|
> > > | Qwen3-8B | 81.85 | 80.46 | 78.82 | 86.16 | 83.92 | 75.56 |
> > > | RedSage-8B-Ins | 85.73 | 84.20 | 84.98 | 89.06 | 86.80 | 80.30 |
> > > | RedSage-8B-DPO | 84.83 | 82.48 | 83.80 | 88.54 | 86.30 | 79.30 |
> > > | Qwen3-32B | 85.40 | 84.08 | 82.32 | 89.00 | 87.60 | 80.40 |
> > > | **GPT-5** | **88.68** | **88.74** | **86.54** | **91.43** | **90.80** | **83.14** |
> > >
> > > ---
> > >
> > > #### **Related Cybersecurity Benchmarks**
> > >
> > > | Model | Mean | CTI-Bench MCQ | CTI-Bench RCM | CyMtc 500 | MMLU CSec | ScBen En | ScEva MCQ | SECURE CWET | SECURE KCV | SECURE MEAT |
> > > |--------|------:|---------------:|---------------:|-----------:|-----------:|-----------:|-----------:|-----------:|-----------:|-----------:|
> > > | Qwen3-8B | 75.71 | 62.76 | 54.00 | 88.60 | 76.00 | 73.26 | 65.46 | 88.11 | 87.42 | 85.75 |
> > > | RedSage-8B-Ins | 81.30 | 70.56 | 76.70 | 89.80 | 78.00 | 79.91 | 72.48 | 91.45 | 81.34 | 91.47 |
> > > | RedSage-8B-DPO | 81.10 | 70.84 | 70.60 | 90.00 | 79.00 | 80.06 | 74.22 | 91.35 | 82.86 | 91.00 |
> > > | Qwen3-32B | 82.31 | 70.04 | 65.60 | 91.80 | 84.00 | 84.23 | 76.23 | 89.46 | 88.72 | 90.06 |
> > > | **GPT-5** | **86.29** | **76.48** | **74.20** | **95.60** | **86.00** | **87.48** | **83.03** | **92.70** | **88.72** | **92.41** |
> > >
> > > ---
> > >
> > > #### **OpenLLM Leaderboard Benchmark (General)**
> > >
> > > | Model | Mean | MMLU | ARC-C | GSM8K | HSwag | TQA | WinoG | IFEval |
> > > |--------|------:|------:|------:|------:|------:|------:|------:|------:|
> > > | Qwen3-8B | 65.92 | 73.59 | 62.54 | 75.66 | 56.70 | 45.23 | 62.51 | 85.21 |
> > > | RedSage-8B-Ins | 73.34 | 77.38 | 69.62 | 86.05 | 79.00 | 47.75 | 73.64 | 79.97 |
> > > | RedSage-8B-DPO | 74.33 | 77.07 | 71.76 | 82.71 | 79.87 | 52.47 | 73.01 | 83.44 |
> > > | Qwen3-32B | 73.17 | 82.11 | 69.28 | 87.49 | 70.93 | 48.17 | 65.98 | 88.26 |
> > > | **GPT-5** | **91.07** | **91.40** | **95.31** | **91.36** | **94.85** | **87.10** | **87.85** | **89.60** |
> > >
> > > ---
> > > **Analysis**
> > >
> > > We find that our **8B models** are comparable to, and in some cases outperform, larger **32B models** on cybersecurity-specific benchmarks. For instance, **RedSage-8B-Ins** slightly outperforms **Qwen3-32B** on our internal MCQ benchmark (+0.23 macro accuracy), while **GPT-5** achieves 88.68 (+4.99 higher).
> > >
> > > On related cybersecurity benchmarks, our post-trained models achieve **81.10–81.30 mean accuracy**, improving by **+5.3–+5.6** over **Qwen3-8B** and remaining close to **Qwen3-32B (82.31, only +1.0–+1.2 higher)**. In contrast, **GPT-5** leads with **86.29 mean accuracy**, about **+5 points higher** than ours.
> > >
> > > For general Open LLM Leaderboard tasks, the performance gap widens, with **GPT-5** reaching **91.07 mean accuracy**. Nonetheless, **RedSage-8B-DPO** achieves **74.33**, slightly higher than **Qwen3-32B (73.17)** due to gains in **HellaSwag**, **TQA**, and **WinoGrande**, which measure commonsense reasoning and factuality. The differences on **MMLU**, **GSM8K**, and **IFEval** are consistent with expectations for smaller models.
> > >
> > > Importantly, our models contain only **8 billion parameters** and can be **deployed locally on consumer-grade GPUs**, while larger models such as **Qwen3-32B** and **GPT-5** require specialized AI infrastructure. **GPT-5**, estimated to have **hundreds of billions to over a trillion parameters** (possibly using a **Mixture-of-Experts architecture**), benefits from far greater capacity in data size and training.
> > >
> > > Despite this scale difference, the relatively small performance gap in cybersecurity tasks highlights the **efficiency and practical value of domain-tuned models** like **RedSage**, emphasizing the **effectiveness and impact of our approach**.

---

### Official Review · Reviewer_mCDa · 2025-11-03

**Soundness:** 3
**Presentation:** 3
**Contribution:** 2
**Rating:** 6
**Confidence:** 3

**Summary:**

This paper curates a corpus of data (11.8B tokens) by filtering FineWeb and perform continued pre-training from Qwen3-8B-Base. The resulting models improve cybersecurity multiple-choice benchmarks. Along the way, the authors also introduce a new benchmark, RedSage-Bench.

**Strengths:**

The paper is well-motivated: to improve a domain-specific capability (cybersecurity), go and curate data for it and fine-tune an existing model.
The paper is fairly well-written and easy to follow.

**Weaknesses:**

There is not really any methodological novelty, given this is similar to other methods like FineWeb-Edu.
I would have liked to see evaluation of larger models (at least 32B) to see how well the methodology transfers to stronger models (one might worry that the gap will shrink).
It would also be good to get a closed model (e.g., GPT-5 or Claude) to get a ceiling for the new benchmark.
The abstract claims that the fine-tuned model improves on OpenLLM leaderboard tasks, but looking at Table 6, it seems like except on GSM8K, the RedSage is worse as would be expected.

**Questions:**

How were the labels for the ModernBERT-base clasisifer obtained? Were there an LLMs that were used?

---

> ### Author Response · Authors · 2025-11-21
> **Reply to Reviewer mCDa: Part1**
>
> We thank the reviewer for the encouraging and insightful comments. Please find our responses to specific queries below.
>
> ---
>
> ### 1. On novelty and similarity to FineWeb-Edu.
>
> Our methodological contribution includes a **fully open, domain-aware pipeline for continual pre-training and post-training** to develop cybersecurity assistant LLM. Our pre-training stage combines large-scale filtered cybersecurity text data (conceptually similar to FineWeb-Edu), with manually curated, high-quality, and categorized cybersecurity datasets.
>
> These curated data are then augmented using our **novel agentic-augmentation** pipeline, which transforms these cybersecurity texts into ∼266K of realistic security dialogues used for model post-training. We also construct categorized cybersecurity benchmarks featuring both multiple-choice questions (MCQs) and open-ended QA tasks, including a tool proficiency category.
>
> To the best of our knowledge, we are the first to present this **end-to-end pipeline for cybersecurity-domain LLM**. Our results demonstrate that it not only boosts performance on cybersecurity benchmarks but also preserves or improves the model’s general knowledge capabilities. While large-scale pre-training on filtered data is an important part of our method, it represents only one element of a broader, integrated RedSage framework.
>
> ---
>
> ### 2. Evaluation of larger models (at least 32B) and how well the methodology transfers to stronger models.
>
> Thank you for the suggestion. Our full fine-tuning of the **8B model** was conducted on **32×A100 64GB GPUs**, requiring about **5.5 days (>4,000 GPU-hours)** for domain-aware pretraining and post-training. Scaling this setup to a **32B model** would require roughly **four times more compute** using the same resources or more GPUs to maintain a similar duration.
>
> Our results show that the **8B RedSage models** already perform comparably to, and in some cases surpass, larger **32B models** on cybersecurity benchmarks. To demonstrate that our **data curation and augmentation pipeline** scales effectively, we conducted a reduced-scope experiment. We fine-tuned **Qwen3-32B** using **QLoRA** on a subset of our data: the **curated RedSage-Seed** dataset (excluding RedSage-Dump) and **50% of RedSage-Conv**.  Even with partial data and lightweight **QLoRA fine-tuning** (≈1% trainable parameters), the **32B model** achieved consistent gains across benchmarks. Training with the full dataset and performing full fine-tuning would likely yield larger improvements, as the training loss was still decreasing at the end of the run. These results confirm that our **RedSage data curation and augmentation methodology** transfers effectively to larger models, reinforcing its **scalability and impact** on cybersecurity LLM development. Tables below summarize the benchmark results.
>
> | **RedSage-MCQ Benchmark** | Macro Acc | Knowledge Gen | Knowledge Frm | Skill Off | Tools CLI | Tools Kali |
> |----------------------------|-----------:|---------------:|---------------:|-----------:|-----------:|------------:|
> | Qwen3-8B | 81.85 | 80.46 | 78.82 | 86.16 | 83.92 | 75.56 |
> | Qwen3-32B | 85.40 | 84.08 | 82.32 | 89.00 | 87.60 | 80.40 |
> | RedSage-8B-Ins | 85.73 | 84.20 | 84.98 | 89.06 | 86.80 | 80.30 |
> | **RedSage-32B-Ins-0.5ep** | **87.53** | **85.68** | **85.04** | **91.46** | **88.76** | **82.78** |
>
> | **Related Cybersecurity Benchmarks** | Mean | CTI-Bench MCQ | CTI-Bench RCM | CyMtc 500 | MMLU CSec | ScBen En | ScEva MCQ | SECURE CWET | SECURE KCV | SECURE MEAT |
> |-------------------------------------|------:|---------------:|---------------:|-----------:|-----------:|-----------:|-----------:|------------:|------------:|------------:|
> | Qwen3-8B | 75.71 | 62.76 | 54.00 | 88.60 | 76.00 | 73.26 | 65.46 | 88.11 | 87.42 | 85.75 |
> | Qwen3-32B | 82.31 | 70.04 | 65.60 | 91.80 | 84.00 | 84.23 | 76.23 | 89.46 | 88.72 | 90.06 |
> | RedSage-8B-Ins | 81.30 | 70.56 | 76.70 | 89.80 | 78.00 | 79.91 | 72.48 | 91.45 | 81.34 | 91.47 |
> | **RedSage-32B-Ins-0.5** | **82.85** | **71.64** | **66.10** | **93.40** | **84.00** | **83.77** | **78.30** | **92.18** | **83.29** | **92.97** |

---

> > ### Author Response · Authors · 2025-11-21
> > **Reply to Reviewer mCDa: Part2**
> >
> > ### 3. Evaluation using closed-source models (e.g., GPT-5 or Claude) to establish a ceiling for the new benchmark.
> >
> > Thank you for the suggestion. We included **GPT-5** in our evaluation to observe how a state-of-the-art closed-source model performs on our new benchmarks and related cybersecurity datasets.
> >
> > Our **RedSage-8B-Ins** and **RedSage-8B-DPO** models achieved **85.73–84.83 macro accuracy**, improving by **+3.88–+2.98** over **Qwen3-8B**, while **GPT-5** scored **88.68**, about **+3 points higher** than our models. On related cybersecurity benchmarks, our models reached **81.30–81.10 mean accuracy**, improving by **+5.3–+5.6** over **Qwen3-8B**, whereas **GPT-5** achieved **86.29**, around **+5 points higher**.
> >
> > Although **GPT-5** surpasses our models, it is estimated to have **hundreds of billions to over a trillion parameters** and possibly uses a **Mixture-of-Experts (MoE) architecture**. In contrast, our **8B RedSage models** can be **deployed locally on consumer-grade GPUs** while maintaining competitive performance. This relatively small performance gap in cybersecurity tasks highlights the **efficiency and impact of domain-tuned models** like RedSage and underscores the **practical significance of our contribution**.
> >
> >
> > **Our Benchmarks**
> >
> > | Model | Macro Acc | Knowledge Gen | Knowledge Frm | Skill Off | Tools CLI | Tools Kali |
> > |--------|-----------:|---------------:|---------------:|-----------:|-----------:|------------:|
> > | Qwen3-8B | 81.85 | 80.46 | 78.82 | 86.16 | 83.92 | 75.56 |
> > | RedSage-8B-Ins | 85.73 | 84.20 | 84.98 | 89.06 | 86.80 | 80.30 |
> > | RedSage-8B-DPO | 84.83 | 82.48 | 83.80 | 88.54 | 86.30 | 79.30 |
> > | **GPT-5** | **88.68** | **88.74** | **86.54** | **91.43** | **90.80** | **83.14** |
> >
> > **Related Cybersecurity Benchmarks**
> >
> > | Model | Mean | CTI-Bench MCQ | CTI-Bench RCM | CyMtc 500 | MMLU CSec | ScBen En | ScEva MCQ | SECURE CWET | SECURE KCV | SECURE MEAT |
> > |--------|------:|---------------:|---------------:|-----------:|-----------:|-----------:|-----------:|-----------:|-----------:|-----------:|
> > | Qwen3-8B | 75.71 | 62.76 | 54.00 | 88.60 | 76.00 | 73.26 | 65.46 | 88.11 | 87.42 | 85.75 |
> > | RedSage-8B-Ins | 81.30 | 70.56 | 76.70 | 89.80 | 78.00 | 79.91 | 72.48 | 91.45 | 81.34 | 91.47 |
> > | RedSage-8B-DPO | 81.10 | 70.84 | 70.60 | 90.00 | 79.00 | 80.06 | 74.22 | 91.35 | 82.86 | 91.00 |
> > | **GPT-5** | **86.29** | **76.48** | **74.20** | **95.60** | **86.00** | **87.48** | **83.03** | **92.70** | **88.72** | **92.41** |
> >
> >
> > ---
> >
> > ### 4.: Clarification on abstract claim versus Table 6 results on OpenLLM benchmarks.
> >
> > In **Table 6** of the OpenLLM leaderboard evaluation, our final fine-tuned models, **RedSage-8B-Ins** and **RedSage-8B-DPO**, achieve the highest overall mean scores of **73.34** and **74.33**, respectively. In comparison, **Qwen3-8B** attains **65.92**, which is **7.42–8.41 points lower**, and best cyber-tuned **Foundation-Sec-8B-Instruct** achieves **69.28**, also **4.06-5.05** below our results.
> >
> > While **RedSage** performs slightly below **Qwen3-8B-Base (70.86)** after **Continued Pretraining (CPT)**, the difference is minimal (≈1%, 69.23–69.58), which is expected for domain-tuned models. Importantly, our model shows minimal degradation during CPT and even **improves general tasks** such as **GSM8K**.
> >
> > Thus, the abstract’s statement that our fine-tuned models improve on OpenLLM leaderboard tasks refers to the **final instruction-tuned variants**, which outperform both base and instruct-tuned baselines. We hypothesize that our **augmented post-training process** strengthens the model’s internal knowledge learned during pretraining, leading to better overall performance across general benchmarks.
> >
> > ---
> >
> > ### Q1: How were the labels for the ModernBERT-base classifier obtained? Were any LLMs used?
> >
> > As detailed in **Appendix A.1 (L822–L830)**, we used the **Cybersecurity Topic Classification dataset** (*Pelofske et al., 2021*), which contains **9.27M labeled training samples** (cybersecurity vs. non-cybersecurity) from **Reddit**, **StackExchange**, and **arXiv**, plus **459K validation samples** from web articles similar to **FineWeb**.
> >
> > Labels were derived from **forum categories**, **topic tags**, and **keyword filtering**. After removing very short texts, the dataset contained **4.62M training samples** and **2,460 validation samples**. The **ModernBERT-base classifier** was trained for **2 epochs** using **Adam**, a **2e-5 learning rate**, and a **10% warmup ratio**, achieving **93.8% precision**, **90.2% recall**, **91.4% F1**, and **97.3% accuracy** on validation.
> >
> > No LLMs were used for labeling, as the source datasets already provided **explicit topic annotations**.

---

### Author Response · Authors · 2025-11-21
**Author Final Response**

Dear Chairs,

Thank you. We have provided comprehensive responses to all reviewer comments in the official section below.

We sincerely thank all reviewers for their positive feedback and insightful comments. The paper was recognized as well-motivated, clearly written, and easy to follow (mCDa); it demonstrates that cybersecurity-curated pre- and post-training significantly improves performance without catastrophic forgetting, and that the curated dataset is valuable for future research (f3PV); and it offers a clear presentation, comprehensive data coverage through CyberFineWeb, RedSage-Seed, and agentic augmentation, with a benchmark that fills a key gap in cybersecurity LLM assessment (QvXR). Reviewers also acknowledged excellent contribution (f3PV, QvXR), excellent presentation (f3PV), and good soundness (mCDa, f3PV, QvXR) scores. We will publicly release code, datasets, and pretrained RedSage models to support the community.

We summarize our main contributions as follows:

- To the best of our knowledge, we present the first fully open, domain-aware pipeline that combines continual pre-training and post-training to build a locally deployable 8B cybersecurity assistant LLM. The resulting models strengthen cybersecurity expertise while preserving and often improving general reasoning and instruction following.

- Our data contribution includes: (1) a large-scale cybersecurity web-filtered corpus for continued pre-training (11.7B tokens); (2) 28.6K manually curated, high-quality cybersecurity documents (149.8M tokens) categorized into knowledge, offensive skills, and tools, plus an additional 459K-document cybersecurity dump (700M tokens); (3) a novel agentic augmentation pipeline that produces 266K multi-turn cybersecurity conversations for supervised fine-tuning; and (4) RedSage-Bench, with 30K MCQs and 240 open-ended, human-verified items covering knowledge, skills, and tool use.

- Our evaluation shows that RedSage consistently outperforms strong baselines on both cybersecurity benchmarks and the Open LLM Leaderboard suite, with gains up to +5.59 points on cybersecurity tasks and +5.05 points on general benchmarks. On several metrics, our 8B models approach or surpass larger models such as Qwen3-32B, underscoring the effectiveness of domain-tuned models.

During the discussion period we incorporated reviewer suggestions and strengthened the paper with additional clarifications, analyses, and experiments:

- **Larger-model scaling and closed-source ceiling (mCDa, f3PV).** We added a 32B scaling experiment (Qwen3-32B with QLoRA on partial RedSage data) in Appendix D.1, which shows consistent gains and demonstrates that our curation and augmentation pipeline transfers to higher-capacity models. We also report GPT-5 and Qwen3-32B results in all major tables, showing that RedSage-8B comes close to, or even exceeds, these larger models on cybersecurity benchmarks, while a clearer gap remains compared to GPT-5 on purely general tasks.

- **Open-ended evaluation, teacher/verifier choice, human verification, decontamination, and agentic benchmark exclusion (f3PV, QvXR).** Section 3.3 and Appendix C.2 now contain more detail on the LLM-as-judge rubric, scoring procedure, human quality control, and prompt templates used. We document our teacher and verifier choices (Llama-3.3-70B and Qwen2.5-72B). All 240 open-ended items are human verified using the same rubric. We also describe a semantic-similarity-based decontamination step that removes training items overly similar to benchmark questions. We explain the exclusion of agentic CTF benchmarks that primarily test interactive tools rather than base LLM ability.

- **Novelty, abstract claim, and classifier labels (mCDa).** We clarify that RedSage goes beyond simple web filtering (e.g., FineWeb-Edu filtering): it provides a full domain-aware pipeline for both continual pretraining and post-training, with newly curated datasets, agentic augmentations, and a dedicated benchmark. We also clarify in the paper and discussion that the abstract’s claim about improvements on Open LLM Leaderboard tasks refers specifically to the final instruction-tuned RedSage models and is directly supported by the reported results. We expand the description in Appendix A.1 to note that ModernBERT classifier labels originate from an existing cybersecurity topic classification dataset rather than LLM labeling.

- **Training time, compute, and presentation improvements (f3PV, QvXR).** Appendix B.3 now contains a training-time and compute breakdown; the full 8B pipeline requires about 5.5 days on 32×A100-64GB (≈4,000 GPU-hours) across CPT, SFT, and DPO. We added a model-derivation diagram as a new Fig. 5 and increased overall figure font sizes to improve readability.

We believe these revisions substantially address the reviewers’ concerns and further clarify the contribution and impact of RedSage as an open, reproducible, and practically useful cybersecurity LLM framework for the community.

---

### Meta-Review · Area_Chair_rBRs · 2026-01-04

**Summary:**

The paper presents an end-to-end pipeline for building a cybersecurity assistant LLM, RedSage, at the 8B scale. Experiments show that RedSage achieves better performance than other LLMs of similar size on cybersecurity knowledge tasks. All reviewers agree that the work is clearly written and provides a substantial practical contribution, especially through the proposed dataset. The main concerns include limited methodological novelty, the limited model scale, insufficient analysis of computational cost, and missing details related to the agentic augmentation process.

**Reviewer Concerns:**

Concerns were addressed in the rebuttal:
1. The concern about evaluation on larger models is partially addressed by fine-tuning Qwen3-32B using the proposed pipeline (Reviewer mCDa).
2. Evaluation against larger/closed-source models is addressed (Reviewers mCDa, f3PV).
3. Requests for clarification and additional methodological details are addressed (Reviewer f3PV).
4. The lack of computational cost analysis is addressed (Reviewers f3PV, QvXR).

Concerns not fully addressed in the rebuttal:
1. The concern about limited methodological novelty remains (Reviewer mCDa).
2. The concern about errors or bias introduced during LLM-generated augmentation remains (Reviewer QvXR).

**Reviewer Scores:**

Reviewers are likely to maintain their scores after the rebuttal, as the remaining concerns about novelty and potential LLM-generated augmentation errors may persist.

---

### Decision · Program_Chairs · 2026-01-26

Accept (Poster)